# Skyrmion Jellyfish in Driven Chiral Magnets

Nina del Ser[1] and Vivek Lohani[1]

[1]*Institute for Theoretical Physics, University of Cologne, 50937 Cologne, Germany*

(Dated: July 20, 2023)

Chiral magnets can host topological particles known as skyrmions, which carry an exactly quantised topological charge $Q = -1$. In the presence of an oscillating magnetic field $\mathbf{B}_1(t)$, a single skyrmion embedded in a ferromagnetic background will start to move with constant velocity $\mathbf{v}_{\text{trans}}$. The mechanism behind this motion is similar to the one used by a jellyfish when it swims through water. We show that the skyrmion's motion is a universal phenomenon, arising in any magnetic system with translational modes. By projecting the equation of motion onto the skyrmion's translational modes and going to quadratic order in $\mathbf{B}_1(t)$, we obtain an analytical expression for $\mathbf{v}_{\text{trans}}$ as a function of the system's linear response. The linear response and consequently $\mathbf{v}_{\text{trans}}$ are influenced by the skyrmion's internal modes and scattering states, as well as by the ferromagnetic background's Kittel mode. The direction and speed of $\mathbf{v}_{\text{trans}}$ can be controlled by changing the polarisation, frequency and phase of the driving field $\mathbf{B}_1(t)$. For systems with small Gilbert damping parameter $\alpha$, we identify two distinct physical mechanisms used by the skyrmion to move. At low driving frequencies, the skyrmion's motion is driven by friction, and $v_{\text{trans}} \sim \alpha$, whereas at higher frequencies above the ferromagnetic gap, the skyrmion moves by magnon emission, and $v_{\text{trans}}$ becomes independent of $\alpha$.

## I. INTRODUCTION

Medusas, commonly known as jellyfish, are remarkable creatures. At over 500 million years of age, which is one hundred times older than Homo sapiens, they are the oldest multi-organ animal on Earth. Jellyfish usually consist of a bell-like structure with tentacles attached to it. While the tentacles serve to stun and catch prey, the jellyfish actually relies on its bell to swim. By periodically relaxing and contracting the bell, the jellyfish generates mini-vortices in the surrounding water which help propel it forward. Over the course of this paper we will show that a magnetic skyrmion, a kind of microscopic particle which occurs naturally in certain classes of magnets, can also move through its environment using a mechanism very similar to that of the jellyfish.

Chiral magnets are predominantly ferromagnetic materials where the inversion symmetry of the crystal lattice is broken. The energy term responsible for this is known as the Dzyaloshinskii-Moriya Interaction (DMI) and originates from weak spin-orbit interactions [1]. Physically the DMI favours neighbouring spins to be perpendicular to each other, thus encouraging twisting in the magnetic texture. Chiral magnets can host different textures including helical and conical phases [2, 3], where the spins wind around a pitch vector $\mathbf{q}$. The reciprocal lattices for these phases carry just one finite momentum $\pm\mathbf{q}$ mode, which also means they are topologically trivial textures. In a small phase pocket near a critical temperature $T_c$, and in the presence of a stabilising external magnetic field, chiral magnets can also host a skyrmion phase, consisting of a hexagonal lattice of topologically quantised magnetic whirls called skyrmions [4]. Single skyrmions can also be created by irradiating ferromagnetic samples with spin-polarised currents using STM tips [5, 6]. Due to its non trivial spatial structure, a single skyrmion has infinitely many $\mathbf{k}$ modes in the Fourier domain, unlike

the helical and conical textures, and additionally carries a finite quantised topological charge $Q = -1$.

Generally, a skyrmion will start to move in the presence of external forces if enough symmetries are broken in the system. One option is to break the translational symmetry, for instance by subjecting the skyrmion to magnetic [7–9] or temperature [10, 11] field gradients, electric or spin currents [12, 13], by driving it with an oscillating magnetic field near a wall [14, 15], or even by firing at it with magnons [16, 17]. Another possibility is to break

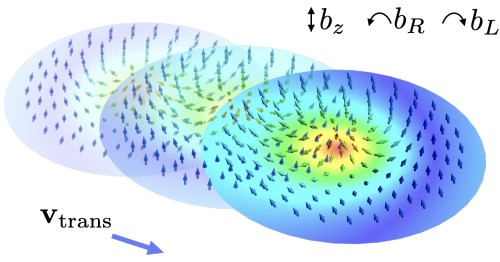

FIG. 1. A two-dimensional Néel skyrmion driven by a tilted oscillating field $\mathbf{B}_1(t) = B_1(\cos(\Omega t), \sin(\Omega t)\cos(\Omega t))$ starts to move in the $xy$-plane with constant velocity $\mathbf{v}_{\text{trans}} \propto B_1^2$. The skyrmion moves through the ferromagnetic bulk as a result of periodic deformations arising at order $\mathcal{O}(B_1)$ from the excitation of its own internal breathing mode, the ferromagnetic background's Kittel mode, and scattering states above the ferromagnetic gap. Here the driving frequency $\Omega = \Omega_{\text{Kit.}}$, which resonantly excites the $m = \pm 1$ momentum sectors, as a result of which the rotational symmetry of the skyrmion gets visibly broken. A jellyfish swims through water using a similar mechanism of cyclic asymmetric contractions and expansions of its bell.

the combined time translation and rotational (around the $\mathbf{e}_z$ axis) symmetries. There are several numerical studies

exploiting this second mechanism, including [18], where a skyrmion was driven with a homogeneous out-of-plane oscillating field in the presence of an in-plane static magnetic field $B_x \mathbf{e}_x$, as well as [14], where there was no $B_x \mathbf{e}_x$, but the driving field was tilted. It is also possible to break the rotational symmetry of the skyrmion by coupling its electric dipole moment to a homogeneous oscillating electric field, see [19]. In this work, we will investigate the same setup as [14], but our results of the skyrmion velocities differ and the full analytical treatment we will provide is novel. We will explain precisely how the skyrmion moves as a result of periodic asymmetric deformations in its shape, see Fig. 1, and why the details of this mechanism share similarities with a jellyfish swimming through water.

When a weakly oscillating spatially homogeneous magnetic field $\mathbf{B}_1(t)$ is applied to a magnetic system, to linear order in perturbation theory magnons oscillating at frequency $\pm\Omega$, where $\Omega$ is the driving frequency, will be excited. Going beyond linear response, at order $\mathcal{O}(B_1^2)$ we would naïvely expect frequency responses at 0 and $2\Omega$. Upon closer inspection, the $\mathcal{O}(B_1^2)$ response in fact also admits a mode which grows *linearly* in time. In a previous work [20], we studied this in the case of driven helical and conical phases, and showed that this linear in $t$ growing mode corresponds precisely to the translational mode of the helix in the $\mathbf{e}_z$-direction. Due to the screw symmetry, the translational motion can equivalently be interpreted as the helix rotating on its axis at constant angular velocity $\Omega_{\text{screw}}$, mimicking an Archimedean screw. In the case of the Archimedean screw, the translational and rotational modes of the magnetic texture coincide, however this is not generally the case. For instance, in a skyrmion lattice the $\mathbf{e}_z$ axis rotational mode is completely separate from any of its translational modes. Driving a skyrmion lattice with an oscillating $\mathbf{B}_1(t)\mathbf{e}_z$ will in fact also activate this rotational mode. Recently, this phenomenon was observed in experiments with femtosecond lasers [21], with large rotation speeds of $2 \times 10^7$–$10^8$ deg s$^{-1}$ achieved near the breathing resonance of the skyrmion lattice. In the present work we will be interested in what happens when we drive a single skyrmion. We will show that in this case, the two translational modes of the skyrmion in the $xy$-plane get activated and the skyrmion acquires a constant velocity $\mathbf{v}_{\text{trans}} = (v_{\text{trans}}^x, v_{\text{trans}}^y, 0)^T$.

In what follows, we will first show that an oscillating spatially homogeneous magnetic field $\mathbf{B}_1(t)$ universally activates the translational mode(s) of chiral magnets. This manifests as an additional $\mathcal{O}(B_1^2)$ force in the Thiele equation, which we derive as a function of the linear order response of the magnet. We then calculate this linear response specifically for a driven single skyrmion and use it to evaluate its second order translational velocity $\mathbf{v}_{\text{trans}}$. We show that the skyrmion has two different ways of achieving maximal $\mathbf{v}_{\text{trans}}$, related to either activating its own internal breathing resonance, or the background ferromagnetic Kittel resonance. This is analogous to a jellyfish swimming by itself versus relying on background

ocean currents to help it swim, except of course that in the case of the skyrmion all the power comes from the driving magnetic field. Just like for the jellyfish, friction via the phenomenological Gilbert damping term plays a fundamental role in our theory for the skyrmion. Paradoxically, we even find that in some driving regimes (below the ferromagnetic gap), the skyrmion actually moves faster in systems with more damping. In the final part of the paper we discuss what happens to $\mathbf{v}_{\text{trans}}$ in various experimentally accessible driving frequency and damping limits, and obtain an excellent match between our analytical predictions and micromagnetic simulations.

## II. MODEL

We consider a chiral magnet whose free energy is given by

$$F = \int \mathrm{d}^3 r \left[ -\frac{J}{2}\hat{\mathbf{M}} \cdot \nabla^2 \hat{\mathbf{M}} + D\hat{\mathbf{M}} \cdot (\boldsymbol{\nabla} \times \hat{\mathbf{M}}) - \mathbf{M} \cdot \mathbf{B}_{\text{ext}} \right],$$ (1)

where $\hat{\mathbf{M}} = \mathbf{M}/M_0$, encodes the unit local magnetisation and we are using spins of fixed length $M_0 = |\mathbf{M}|$. For simplicity we neglect dipole-dipole interactions. We consider an external magnetic field consisting of a static component $\mathbf{B}_0$ and a dynamic driving component $\mathbf{B}_1(t)$,

$$\mathbf{B}_{\text{ext}} = \mathbf{B}_0 + \epsilon \mathbf{B}_1(t), \qquad \mathbf{B}_0 = (0, 0, B_0)^T,$$
$$\mathbf{B}_1(t) = (B_\perp^x \cos(\Omega t), B_\perp^y \sin(\Omega t), B^z \cos(\Omega t + \delta))^T.$$ (2)

We will concentrate on the weak driving limit $B_1/B_0 \ll 1$, using $\epsilon$ as a book-keeping parameter in perturbation theory in the later sections. In the absence of driving, $\mathbf{B}_1 = 0$, and close to the critical temperature $T_c$, the ground state of Eq. (1) is a hexagonal skyrmion lattice. If the stabilising $\mathbf{B}_0$ field is large enough, a single skyrmion also becomes a stable excitation of the ferromagnetic phase [17]. This is the texture we will be interested in here. Due to its rotational symmetry, a single skyrmion is most naturally parametrised using polar coordinates,

$$\hat{\mathbf{M}}^{(0)} = \begin{pmatrix} \sin\left(\theta_0(r)\right)\cos(\chi + h) \\ \sin\left(\theta_0(r)\right)\sin(\chi + h) \\ \cos\left(\theta_0(r)\right) \end{pmatrix},$$ (3)

where $r = \sqrt{x^2 + y^2}$ is the radial coordinate and $\chi = \arctan(y/x)$ is the polar angle. To model the ferromagnetic background, where $\hat{\mathbf{M}}^{(0)} = \mathbf{e}_z$, we set the boundary condition $\lim_{r \gg r_0} \theta_0(r) = 0$, where $r_0$ is the radius of the skyrmion. In Eq. (3), $h$ is short for helicity, a parameter determined by the type of DMI used in the model. In Eq. (1), we used the bulk form of DMI, in which case the free energy is minimised when $h = \pi/2$, giving rise to Bloch skyrmions. We could instead have used interfacial DMI, given by $D\left(\hat{M}_z \nabla \cdot \hat{\mathbf{M}} - (\hat{\mathbf{M}} \cdot \nabla)\hat{M}_z\right)$. In that case, $h = 0$ minimises $F$, giving rise to Néel skyrmions.

Eq. (3) is translationally invariant in the $\mathbf{e}_z$-direction, so that in 3D bulk materials the texture forms skyrmion tubes.

To determine the radial dependence of the angle $\theta_0(r)$, we substitute the skyrmion ansatz Eq. (3) into Eq. (1) and solve the Euler-Lagrange equation

$$\frac{\delta F}{\delta \theta_0} = \frac{1}{r}\frac{d}{dr}\left(r\frac{\delta F}{\delta \theta_0'}\right), \tag{4}$$

where $\theta_0' = \frac{d\theta_0}{dr}$ and the extra $r$-factor on the right side comes from using polar coordinates. For notational clarity, we now switch to dimensionless radial units $\tilde{r} = (D/J)\,r$ and reduced magnetic field units $b_0 = (M_0 J/D^2)\,B_0$. Eq. (4) then evaluates to

$$\theta_0'' + \frac{1}{r}\theta_0' - \frac{\sin(\theta_0)}{r^2}\left(b_0 r^2 - 2r\sin(\theta_0) + \cos(\theta_0)\right) = 0, \tag{5}$$

with boundary conditions $\theta_0(r = 0) = \pi$ and $\theta_0(r = \infty) = 0$, where for simplicity we dropped the tilde on $\tilde{r}$. Eq. (5) is identical for both Bloch and Néel skyrmions, so conveniently only needs to be solved once. The asymptotic limits $r \to 0$ and $r \to \infty$ admit the leading order solutions

$$\theta_0(r \to 0) \approx \pi + \theta_0'(0)r,$$
$$\theta_0(r \gg r_0) \approx \frac{Ae^{-\sqrt{b_0}r}}{\sqrt{r}}. \tag{6}$$

In the intermediate regime $0 < r < r_0$, $\theta_0(r)$ has no known analytical solution, so we solve for it numerically using a shooting method implemented on julia [17, 22, 23]. Using the initial condition $\theta_0(0) = \pi$, we vary $\theta_0'(0)$ until $\theta_0(r \gg r_0)$ decays to zero. For large enough $r$, we match this numerical solution to the analytical asymptotic Eq. (6), giving us a numerical value for $A$. The resulting profile $\theta_0(r)$ depends only on the static external field: the larger $b_0$ is, the faster $\theta_0(r)$ decays to zero, and therefore the smaller the skyrmion. For simplicity, we set $b_0 = 1$ in the plots and numerical simulations shown in later parts of the text, safe in the knowledge that changing the value of $b_0$ will shift the resonances in the frequency spectrum but not introduce any other new features.

## III. ACTIVATING THE TRANSLATIONAL MODE(S)

We now turn on the driving field $\mathbf{B}_1(t)$. The magnetic texture evolves in time according to the Landau-Lifshitz-Gilbert (LLG) equation,

$$\dot{\mathbf{M}} = \gamma\mathbf{M}\times\mathbf{B}_{\text{eff}} - \frac{\gamma}{|\gamma|}\alpha\hat{\mathbf{M}}\times\dot{\mathbf{M}}, \tag{7}$$

where $\mathbf{B}_{\text{eff}} = -\frac{\delta F[\mathbf{M}]}{\delta \mathbf{M}}$ is the effective magnetic field, calculated by taking the functional derivative of the free

energy, Eq. (1). The term proportional to $\alpha$ is a phenomenological damping term and $\gamma = qg/(2m)$ is the gyromagnetic ratio. In our convention, $\gamma_e = -|e|g/(2m_e)$ is negative for an electron with charge $-|e|$, mass $m_e$ and $g$-factor $g$, which is also the case in most magnetic systems. Nevertheless we have included the prefactor $\gamma/|\gamma|$ to ensure that Eq. (7) and all ensuing expressions are also valid for systems with positive $\gamma$. Note that Eq. (7) is the general equation of motion for *any* magnetic texture, not just the skyrmion. Consequently, the phenomenon we are deriving is universal.

As $\mathbf{B}_1(t)$ is assumed to be weak, we can expand the magnetisation perturbatively,

$$\hat{\mathbf{M}}(\mathbf{r},t) = \hat{\mathbf{M}}^{(0)}(\mathbf{r}) + \epsilon\mathbf{M}^{(1)}(\mathbf{r},t) + \epsilon^2\mathbf{M}^{(2)}(\mathbf{r},t) + \mathcal{O}(\epsilon^3), \tag{8}$$

where $\epsilon$ is the same book-keeping parameter we introduced in Eq. (2) to keep track of the powers of $B_1$. Linear response dictates that $\mathbf{M}^{(1)}(\mathbf{r},t)$ should exactly mimic the frequency dependence of the drive. For the type of monochromatic driving field we are using in Eq. (2), this implies that $\mathbf{M}^{(1)}(\mathbf{r},t)$ must also be purely oscillatory, with frequency components $\pm\Omega$. Thus any net translational motion, which is linear in $t$, can only enter at $\mathcal{O}(\epsilon^2)$ or above. A more accurate ansatz describing a magnetic texture which is being translated in time at a constant $\mathcal{O}(\epsilon^2)$ velocity $\mathbf{v}_{\text{trans}}$ is given by

$$\hat{\mathbf{M}}(\mathbf{r},t) = \hat{\mathbf{M}}^{(0)}\left(\mathbf{r} - \epsilon^2\mathbf{v}_{\text{trans}}t\right) + \epsilon\mathbf{M}^{(1)}_{\text{osc.}}(\mathbf{r},t)$$
$$+ \epsilon^2\left(\mathbf{M}^{(2)}_{\text{osc.}}(\mathbf{r},t) + \mathbf{M}^{(2)}_{\text{stat.}}(\mathbf{r})\right) + \mathcal{O}(\epsilon^3), \tag{9}$$

where $\mathbf{M}^{(1)}_{\text{osc.}} \sim e^{\pm i\Omega t}$, $\hat{\mathbf{M}}^{(2)}_{\text{osc.}} \sim e^{\pm 2i\Omega t}$, while $\hat{\mathbf{M}}^{(2)}_{\text{stat.}} \sim e^{0i\Omega t}$ is time-independent. Comparing Eq. (9) and Eq. (8), we see that Eq. (8) is only a valid approximation at short times, as the $\mathcal{O}(\epsilon^2)$ terms grow linearly in $t$ and will eventually get larger than the $\mathcal{O}(\epsilon^0, \epsilon^1)$ terms. We can get rid of this problem by differentiating everything once with respect to time. Taking the time derivatives of Eq. (8) and Eq. (9) and collecting the $\mathcal{O}(\epsilon^1)$ and $\mathcal{O}(\epsilon^2)$ terms separately, we have

$$\dot{\mathbf{M}}^{(1)}(\mathbf{r},t) = \dot{\mathbf{M}}^{(1)}_{\text{osc.}}(\mathbf{r},t), \tag{10}$$
$$\dot{\mathbf{M}}^{(2)}(\mathbf{r},t) = -(\mathbf{v}_{\text{trans}}\cdot\boldsymbol{\nabla})\hat{\mathbf{M}}^{(0)}(\mathbf{r}) + \dot{\mathbf{M}}^{(2)}_{\text{osc.}}(\mathbf{r},t). \tag{11}$$

Substituting Eq. (11) into Eq. (7), operating on it with $\int d^3r\nabla_i\hat{\mathbf{M}}\cdot(\hat{\mathbf{M}}\times)$ and time averaging over one period of oscillation, $T = 2\pi/\Omega$, see App. B.1 for technical details, we obtain a Thiele-inspired equation for $\mathbf{v}_{\text{trans}}$,

$$-\text{sgn}(\gamma)\mathbf{G}\times\mathbf{v}_{\text{trans}} + \alpha\mathcal{D}\mathbf{v}_{\text{trans}} = \mathbf{F}_{\text{trans}}[\hat{\mathbf{M}}^{(0)},\mathbf{M}^{(1)}], \tag{12}$$

where the gyrocoupling vector $\mathbf{G}$ and the dissipation matrix $\mathcal{D}$ are defined in the usual way as functions of the static texture $\hat{\mathbf{M}}^{(0)}$,

$$G_\alpha = \frac{1}{2}\epsilon_{\alpha\beta\gamma}\int d^3r\,\hat{\mathbf{M}}^{(0)}\cdot\left(\nabla_\beta\hat{\mathbf{M}}^{(0)}\times\nabla_\gamma\hat{\mathbf{M}}^{(0)}\right),$$
$$\mathcal{D}_{\alpha\beta} = \int d^3r\,\nabla_\alpha\hat{\mathbf{M}}^{(0)}\cdot\nabla_\beta\hat{\mathbf{M}}^{(0)}, \tag{13}$$

and the second order force is given by

$$F_{\text{trans.}}^i = \left\langle -\operatorname{sgn}(\gamma) \int \mathrm{d}^3 r\, \hat{\mathbf{M}}^{(0)} \cdot \left( \dot{\mathbf{M}}^{(1)} \times \nabla_i \mathbf{M}^{(1)} \right) \right.$$
$$\left. + \alpha \int \mathrm{d}^3 r\, \dot{\mathbf{M}}^{(1)} \cdot \nabla_i \mathbf{M}^{(1)} \right\rangle_t, \tag{14}$$

where $\langle \ldots \rangle_t$ denotes time averaging over one period of oscillation $T = 2\pi/\Omega$. Note that Eq. (12) is a completely general Thiele-inspired equation, valid for *any* magnetic texture driven by a homogeneous oscillating magnetic field. Conveniently, the effective force $\mathbf{F}_{\text{trans}}$ only depends on the static texture $\hat{\mathbf{M}}^{(0)}$ and the linear response $\mathbf{M}^{(1)}$ — this means we don't need to know anything about the oscillatory and static responses $\mathbf{M}_{\text{osc.}}^{(2)}$, $\mathbf{M}_{\text{stat.}}^{(2)}$ to calculate $\mathbf{v}_{\text{trans}}$. Remarkably, there is also no trace of $\mathbf{B}_{\text{eff}}$, which disappeared from the equation when we integrated over space due to the translational symmetry of the free energy, see App. B.1. There is an alternative (longer) way to derive Eq. (12), which involves writing down a continuity equation in terms of the stress energy tensor of the system, see App. C. This approach is useful for investigating the momentum and current densities of the magnons excited by the driving and will be exploited in Sec. V.

The presence of the $\nabla_i \mathbf{M}^{(1)}$ terms in Eq. (14) implies that we need some spatial modulation in $\mathbf{M}^{(1)}$ if we want to activate the translational modes of the texture. Immediately, we can conclude that $\mathbf{v}_{\text{trans}} = 0$ for a ferromagnet, as driving a ferromagnet with a homogeneous $\mathbf{B}_1(t)$ will generate a spatially homogeneous $\mathbf{M}^{(1)}(t)$. The next simplest magnetic textures to consider are the helical and conical phases of chiral magnets. If we set up our axes such that the helical pitch vector $\mathbf{q} = q\mathbf{e}_z$, the helical and conical phases are translationally invariant in the $xy$-plane, but spatially modulated in the $\mathbf{e}_z$-direction. Driving a helical or conical phase generates magnons travelling in the $\pm\mathbf{e}_z$-direction at linear order, $\mathbf{M}^{(1)} \sim e^{i(qz \pm \omega t)}$ [20]. Substituting this into Eq. (14), we immediately notice that the $\hat{\mathbf{M}}^{(0)} \cdot (\dot{\mathbf{M}}^{(1)} \times \nabla_z \mathbf{M}^{(1)})$ term vanishes, as $\dot{\mathbf{M}}^{(1)} \parallel \nabla_z \mathbf{M}^{(1)}$. This leaves only the two dissipation terms in Eq. (12), and the common factor $\alpha$ cancels to give a velocity

$$\mathbf{v}_{\text{trans}} = \mathbf{e}_z \frac{\int_0^\lambda dz\, \left\langle \dot{\mathbf{M}}^{(1)} \cdot \nabla_z \mathbf{M}^{(1)} \right\rangle_t}{\int_0^\lambda dz\, |\nabla_z \hat{\mathbf{M}}^{(0)}|^2}, \tag{15}$$

where $\lambda = 2\pi/q$ is the helix wavelength. The translational velocity $v_{\text{trans}}$ can equivalently be interpreted as a rotational angular velocity $\mathbf{\Omega}_{\text{screw}} = q v_{\text{trans}} \mathbf{e}_z$ due to the screw symmetry of the helical and conical phases. The resulting magnetisation dynamics is reminiscent of a rotating Archimedean screw, see also [20] for a video and additional details. Notice that in Eq. (15), we used the discrete lattice symmetry of the helical and conical phases to integrate only over one helix winding $\lambda$, rather than the whole volume of the magnet. This trick can generically be used in other repeating textures, such as

the skyrmion lattice, but will not work in systems lacking lattice symmetry, such as a single skyrmion.

Despite the multiple qualities of Eq. (14), using it to calculate $\mathbf{v}_{\text{trans}}$ doesn't always work in practice. This is due to problems of convergence which arise in the integration step during the calculation of $\mathbf{F}^{\text{trans}}$. If there is no underlying lattice symmetry and the linear response $\mathbf{M}^{(1)}$ decays very slowly over space, the integration domain needs to be huge, making a numerical implementation impractical. Luckily, there is an alternative way of calculating the force which bypasses this difficulty. By projecting $\nabla_i \hat{\mathbf{M}}^{(0)} \cdot (\hat{\mathbf{M}}^{(0)} \times$ onto Eq. (7), see App. B.2 for details, we again obtain the Thiele equation Eq. (12), but the force is calculated in a different way,

$$\tilde{F}_{\text{trans}}^i = |\gamma| \left\langle \int \mathrm{d}^3 r \left[ \left( \nabla_i \hat{\mathbf{M}}^{(0)} \cdot \mathbf{M}^{(1)} \right) \left( \hat{\mathbf{M}}^{(0)} \cdot \mathbf{B}_{\text{eff}}^{(1)} \right) \right. \right.$$
$$\left. \left. + \frac{1}{2} \left( \mathbf{M}^{(1)} \cdot \mathbf{M}^{(1)} \right) \nabla_i \left( \hat{\mathbf{M}}^{(0)} \cdot \mathbf{B}_{\text{eff}}^{(0)} \right) \right] \right\rangle_t. \tag{16}$$

Eq. (16) is not valid in general for any magnetic texture as it requires some surface terms vanishing at infinity, see App. B.2. However, it is *always* valid if $\nabla_i \mathbf{M}^{(0)}$, which parametrises the texture's translational modes, is bounded. This is the case for example for a single skyrmion, where as we leave the skyrmion and enter the spatially homogeneous ferromagnetic bulk, $\nabla_i \mathbf{M}^{(0)} \to 0$. If this condition is fulfilled, then the integrands contained in Eq. (16) are also always bounded, as a bounded $\nabla_i \mathbf{M}^{(0)}$ also implies a bounded $\nabla_i \mathbf{B}_{\text{eff}}^{(0)}$. Hence, irrespective of the behaviour of $\mathbf{M}^{(1)}$, the integrands in Eq. (16) are always bounded by $\nabla_i \mathbf{M}^{(0)}$. This makes a practical numerical implementation of the integral achievable. The price to pay when using this method compared to Eq. (14) is that one also needs to calculate the $\mathbf{B}_{\text{eff}}^{(1)}$ terms — overall, the integrand in Eq. (16) is therefore a much more complicated object. Also, investigating what happens to the force in various limiting values of $\alpha, \Omega$ is very challenging using Eq. (16), but easy to do using Eq. (14).

When using the effective Thiele equation Eq. (12), it is important to be aware of the requirements for its validity. The main assumptions used in the derivation of this equation are i), that the system has already reached its steady state (meaning all transients generated in the response at the start have by now decayed) and ii), that we are working on time scales much larger than the other characteristic time scales of the system, including the driving frequency $\Omega$. Effectively, we are "integrating out" the short-timescale fluctuations in favour of calculating just the $\mathcal{O}(\omega^0)$ drift of the magnetic texture. We have also not included any effects caused by inertia terms [24, 25], expected to arise as a consequence of the periodic deformations of the driven texture. This is because in the steady-state limit we are considering, $\mathbf{v}_{\text{trans}} = \dot{\mathbf{R}} = \text{const.}$, so that any contribution from a mass term would vanish, $m\ddot{\mathbf{R}} = 0$. Under these conditions, we will use both Eq. (14) and Eq. (16) to calculate and understand $\mathbf{v}_{\text{trans}}$ for a single skyrmion in Sec. V. To do this, we first need

the linear response $\mathbf{M}^{(1)}$, which we calculate in the next section.

## IV. LINEAR RESPONSE FOR THE DRIVEN SKYRMION

The goal of this section is to solve Eq. (7) to order $\mathcal{O}(\epsilon)$ to obtain the linear response $\mathbf{M}^{(1)}$. Instead of doing this directly, we will first rewrite our problem in the language of bosonic fields, inspired by (but not identical to) the Holstein-Primakoff approach for quantum spins. Our approach will also include the effects of the phenomenological damping $\alpha$, which is not as evident to describe using quantum Hamiltonians. We begin by parametrising the magnetisation as

$$\mathbf{M} = M_0 \left( \mathbf{e}_3(1 - a^*a) + \sqrt{1 - \frac{a^*a}{2}} \left( \mathbf{e}_- a + \mathbf{e}_+ a^* \right) \right),$$
(17)

where $a(\mathbf{r}, t), a^*(\mathbf{r}, t)$ are complex time- and space- dependent fields. Note that, contrary to the quantum case, the ordering of $a^*$ and $a$ is unimportant here, as they are complex numbers rather than operators. In the coordinate system we use, $\mathbf{e}_3$ is parallel to the zeroth order magnetisation $\hat{\mathbf{M}}^{(0)}$ given in Eq. (3), while $\mathbf{e}_\pm$ span the plane perpendicular to $\hat{\mathbf{M}}^{(0)}$,

$$\mathbf{e}_3 = \hat{\mathbf{M}}^{(0)}, \quad \mathbf{e}_\mp = \frac{1}{\sqrt{2}} \begin{pmatrix} \cos(\theta_0)\cos(\phi) \pm i\sin(\phi) \\ \cos(\theta_0)\sin(\phi) \mp i\cos(\phi) \\ -\sin(\theta_0) \end{pmatrix},$$
(18)

with $\phi = \chi + h$. Using Eq. (18), it can be checked that the important property $|\mathbf{M}| = \sqrt{\mathbf{M} \cdot \mathbf{M}} = M_0$ is preserved by the parametrisation introduced in Eq. (17). By imposing the Poisson bracket $\{a(\mathbf{r}), a^*(\mathbf{r}')\} = \delta(\mathbf{r} - \mathbf{r}')$, we further ensure that

$$\{\hat{M}_i(\mathbf{r}), \hat{M}_j(\mathbf{r}')\} = i\epsilon_{ijk}\hat{M}_k\delta(\mathbf{r} - \mathbf{r}')$$
(19)

to *all* orders of $a, a^*$. Eq. (19) implies that $i\{F, \hat{\mathbf{M}}(\mathbf{r})\} = \mathbf{M} \times \mathbf{B}_{\text{ext}}$, letting us rewrite Eq. (7) as

$$\text{sgn}(\gamma)\dot{\hat{\mathbf{M}}} = i\{F, \hat{\mathbf{M}}\} - \alpha\hat{\mathbf{M}} \times \dot{\hat{\mathbf{M}}},$$
(20)

where we used dimensionless time units defined by $\tilde{t} = D^2|\gamma|/(JM_0)t$, with corresponding dimensionless frequency $\omega = JM_0/(D^2|\gamma|)\Omega$, rescaled the free energy $\tilde{F} = (D/J^2)F$, and then immediately dropped the tildes on $\tilde{t}, \tilde{F}$ to make the formulas neater. Eq. (20) resembles Hamilton's equation of motion, familiar to us from classical mechanics, but additionally includes the phenomenological damping $\alpha$. Projecting Eq. (20) onto $\mathbf{e}_\pm$, we obtain non-linear equations of motion for $a$ and $a^*$, respectively, which are listed in Eq. (D1). We can expand

$a$ and $a^*$ perturbatively in powers of $\epsilon$,

$$a = \epsilon a^{(1)} + \epsilon^2 a^{(2)} + \mathcal{O}(\epsilon^3),$$
$$a^* = \epsilon a^{*(1)} + \epsilon^2 a^{*(2)} + \mathcal{O}(\epsilon^3).$$
(21)

Notice that $a, a^*$ is to lowest order $\mathcal{O}(\epsilon^1)$ and *not* $\mathcal{O}(\epsilon^0)$, unlike the expansion for $\hat{\mathbf{M}}$ in Eq. (8). This is because by definition, $a$ and $a^*$ vanish when the driving field is turned off, $\mathbf{B}_1(t) = 0$. Within our perturbative scheme, it is sufficient to solve Eq. (D1) to order $\mathcal{O}(\epsilon^1)$, which means we only need to retain those terms which are at most linear in $a, a^*$. This leads to the following linearised equations

$$(\text{sgn}(\gamma) + i\alpha)\dot{a} = i\left\{F_{\text{drive}}^{(1)} + F_{\text{no drive}}^{(2)}, a\right\},$$
$$(\text{sgn}(\gamma) - i\alpha)\dot{a}^* = i\left\{F_{\text{drive}}^{(1)} + F_{\text{no drive}}^{(2)}, a^*\right\},$$
(22)

where $F_{\text{drive}}^{(1)}$ and $F_{\text{no drive}}^{(2)}$, refer to the $\mathcal{O}(a^1)$ and $\mathcal{O}(a^2)$ contributions in the free energy given in Eq. (E2) and (E4), respectively. In our notation, $F_{\text{drive}}^{(n)}$ refers to terms which are explicitly dependent on the external driving field, while $F_{\text{no drive}}^{(n)}$ refers to all the other free energy terms. After the Poisson bracket operation in Eq. (22), there will be $\mathcal{O}(a^1)$ terms and $\mathcal{O}(a^0)$ terms coming from $\{F_{\text{no drive}}^{(2)}, a\}$ and $\{F_{\text{drive}}^{(1)}, a\}$, respectively. The $\mathcal{O}(a^0)$ terms coming from $\{F_{\text{drive}}^{(1)}, a\}$ provide a driving force $f$, given in Eq. (F1), which carries only three azimuthal angle Fourier components $e^{im\chi}$: $m = -1, 0$ and 1. We use the Fourier convention

$$a(r, \chi) = \sum_m e^{im\chi} a_m(r),$$
$$a^*(r, \chi) = \sum_m e^{im\chi} a^*_{-m}(r),$$
(23)
$$f(r, \chi) = \sum_m e^{im\chi} f_m(r),$$

and accompanying Poisson bracket,

$$\{a_m(r), a^*_{m'}(r')\} = \frac{1}{2\pi r}\delta_{mm'}\delta(r - r')$$
(24)

(note the factor of $1/r$, a consequence of using polar coordinates!), to write Eq. (22) as a single matrix equation

$$i\left(\text{sgn}(\gamma) + i\alpha\sigma^z\right)\begin{pmatrix} \dot{a}_m^{(1)} \\ \dot{a}_{-m}^{*(1)} \end{pmatrix} = \sigma^z H_m \begin{pmatrix} a_m^{(1)} \\ a_{-m}^{*(1)} \end{pmatrix} + \begin{pmatrix} f_m \\ -f^*_{-m} \end{pmatrix},$$
(25)

where $H_m$ is a $2 \times 2$ matrix defined in Eq. (F2) and

$$f_{-1} = -\frac{1}{4\sqrt{2}}\left(\cos(\theta_0) + 1\right)\left(b_R e^{-i(h-\omega t)} + b_L e^{-i(h+\omega t)}\right)$$

$$f_0 = \frac{1}{2\sqrt{2}}b_z \sin(\theta_0)\left(e^{i(\omega t+\delta)} + e^{-i(\omega t+\delta)}\right)$$
(26)

$$f_1 = -\frac{1}{4\sqrt{2}}\left(\cos(\theta_0) - 1\right)\left(b_R e^{i(h-\omega t)} + b_L e^{i(h+\omega t)}\right).$$

As we are only interested in solutions which oscillate in time at frequency $\pm\omega$, we need only solve Eq. (25) for the values of $m$ where $f_m$ is non-zero. This means we only need to solve the three cases $m = -1, 0$ and $+1$. Inserting the ansatz

$$\begin{pmatrix} a_m^{(1)}(t) \\ a_{-m}^{*(1)}(t) \end{pmatrix} = \begin{pmatrix} a_{m,+\omega}^{(1)} \\ a_{-m,-\omega}^{*(1)} \end{pmatrix} e^{i\omega t} + \begin{pmatrix} a_{m,-\omega}^{(1)} \\ a_{-m,\omega}^{*(1)} \end{pmatrix} e^{-i\omega t},$$

$$f_m = f_{m,+\omega} e^{i\omega t} + f_{m,-\omega} e^{-i\omega t}, \qquad (27)$$

where $a_{m,\pm\omega}^{(1)}$, $a_{-m,\mp\omega}^{*(1)}$, $f_{m,\pm\omega}$ are complex fields which only depend on $r$, into Eq. (25), and collecting the coefficients of the $e^{\pm i\omega t}$ terms, we obtain the time-independent matrix equation

$$\mp\omega \left(\mathrm{sgn}(\gamma) + i\alpha\sigma^z\right) \begin{pmatrix} a_{m,\pm\omega}^{(1)} \\ a_{-m,\mp\omega}^{*(1)} \end{pmatrix} = \sigma^z H_m \begin{pmatrix} a_{m,\pm\omega}^{(1)} \\ a_{-m,\mp\omega}^{*(1)} \end{pmatrix}$$

$$+ \begin{pmatrix} f_{m,\pm\omega} \\ -f_{-m,\mp\omega}^* \end{pmatrix}. \qquad (28)$$

We will solve Eq. (28) by expanding $a_{m,\pm\omega}^{(1)}$, $a_{-m,\mp\omega}^{(1)}$ into the eigenbasis of $\sigma^z H_m$. Sec. IV.1 explains how to obtain this eigenbasis including the effects of the Gilbert damping $\alpha$.

### IV.1. Damped eigenbasis of $\sigma^z H_m$

Using the notation $|m, k, i\rangle$ and $E_{m,k,i}$ to designate the eigenvectors and eigenvalues respectively, the eigenvalue equation reads

$$E_{m,k,i}\left(\mathrm{sgn}(\gamma) + i\alpha\sigma^z\right)|m, k, i\rangle = \sigma^z H_m |m, k, i\rangle, \quad (29)$$

where $k$ parametrises the energy and $i = \{1, 2\}$ labels which of the two eigenvectors of $\sigma^z H_m$ we are referring to. Using the property $\sigma^x H_m \sigma^x = H_{-m}$, it can be shown that if $|m, k, i\rangle$ is an eigenvector of $\sigma^z H_m$ with eigenvalue $E_{m,k,i}$, then $\sigma^x |m, k, i\rangle^*$ is also an eigenvector of $\sigma^z H_{-m}$ with eigenvalue $-E_{m,k,i}^*$, see App. 1 for the derivation. This means that $|-m, k, 2\rangle = \sigma^x |m, k, 1\rangle^*$ and $E_{-m,k,2} = -E_{m,k,1}^*$, and we can write everything in terms of only the $i = 1$ eigenvector and eigenvalue, suppressing the $i$-index in the expressions that follow.

Due to the presence of damping $\alpha$ in Eq. (29), $|m, k\rangle$ and $E_{m,k}$ will in general be complex. Physically, $\mathrm{Re}(E_{m,k})$ corresponds to the frequency of the spin wave while $\mathrm{Im}(E_{m,k})$ quantifies its damping. Assuming that $\alpha \ll 1$, we can expand everything perturbatively in powers of $\alpha$,

$$|m, k\rangle = |m, k^{(0)}\rangle + i\alpha\,\mathrm{sgn}(\gamma) |m, k^{(1)}\rangle + \mathcal{O}(\alpha^2),$$

$$E_{m,k} = \mathrm{sgn}(\gamma)\epsilon_{m,k}^{(0)} - i\alpha\epsilon_{m,k}^{(1)} + \mathcal{O}(\alpha^2). \qquad (30)$$

Stopping at linear order in $\alpha$ in Eq. (30) is sufficient to take into account the leading order effects of damping,

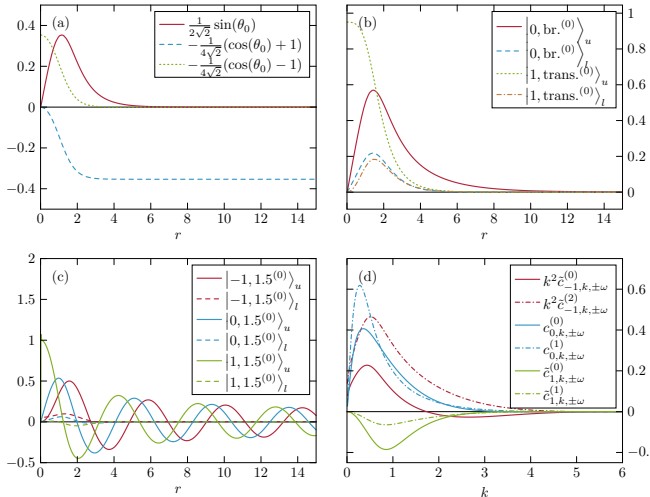

FIG. 2. a) $r$-dependent factors of $f_0$, $f_{\pm 1}$. While the factors in $f_0$ and $f_1$ are bounded and decay as $r \gg r_0 \sim 6$, the factor in $f_{-1}$ tends to a finite constant value $-1/(2\sqrt{2})$ as $r \to \infty$. b) Bound modes of the skyrmion. Only the $m = 0$ breathing mode and $m = \pm 1$ translational modes are excited by spatially homogeneous driving. "$u$" and "$l$" stand for "upper" and "lower" component of the eigenvector, respectively. c) Scattering states of the skyrmion. Here we show one example of a scattering state with $m = 0, \pm 1$ and $k$ set to 1.5. A continuum of such states with $k > 0$ is excited by homogeneous driving. d) Scattering state coefficients $c_{0,k,\pm\omega}^{(0,1)}$, $\tilde{c}_{\pm 1, k, \pm\omega}^{(0,1)}$ for $b_z, b_R, b_L = 1$ and $\delta, h = 0$, plotted as a function of $k$.

thus we will only discuss how to obtain the $\mathcal{O}(\alpha^0)$ and $\mathcal{O}(\alpha^1)$ eigenvectors and eigenvalues. The problem of calculating $|m, k^{(0)}\rangle$, $\epsilon_{m,k}^{(0)}$ has been solved before, see [17] for a detailed account. Very briefly, it involves using the shooting method with some initial conditions at $r = 0$, and imposing that the lower component of $|m, k\rangle$ decays to zero for $r \gg r_0$. The eigenspectrum consists of discrete bound modes with $\epsilon_m^{(0)} < b_0$ and a continuum of scattering states with $\epsilon_m^{(0)} > b_0$. For $m = 0$, there is a single bound mode known as the breathing mode, which we label $\left|0, \mathrm{br.}^{(0)}\right\rangle$, with energy $\epsilon_{\mathrm{br.}}^{(0)} = 0.839$ for $b_0 = 1$. For $m = 1$, there is also a a single bound mode, which happens to be one of the skyrmion's two translational modes (the other one being in the $m = -1$ sector), with energy $\epsilon_{\mathrm{trans}}^{(0)} = 0$. This translational mode is known analytically,

$$\left|m = 1, \mathrm{trans.}^{(0)}\right\rangle = \frac{1}{2\sqrt{2}} \begin{pmatrix} \theta_0' - \frac{\sin(\theta_0)}{r} \\ \theta_0' + \frac{\sin(\theta_0)}{r} \end{pmatrix}, \qquad (31)$$

and can be used to obtain the other translational mode via $\left|m = -1, \mathrm{trans.}^{(0)}\right\rangle = \sigma^x \left|m = 1, \mathrm{trans.}^{(0)}\right\rangle$. The upper and lower components of $\left|0, \mathrm{br.}^{(0)}\right\rangle$ and $\left|m = 1, \mathrm{trans.}^{(0)}\right\rangle$ are plotted as a function of $r$ in Fig. 2(b) — note how they are all confined to the skyrmion radius $r_0 \simeq 6$. For the scattering states

$\left| m, k^{(0)} \right\rangle$, the energy is independent of $m$, and exists on a continuum parametrised by $\epsilon_{m,k} = b_0 + k^2$. We show some examples of the scattering states from the $m = 0, \pm 1$ sectors in Fig. 2(c). Note that, unlike the bound modes, the upper components are *not* confined to the skyrmion radius but instead penetrate far into the ferromagnetic bulk. Far away from the skyrmion and after a few oscillations, the behaviour of the upper components can be described analytically by Bessel functions,

$$\lim_{\substack{r \gg r_0, \\ 2\pi/k}} \left| m, k^{(0)} \right\rangle_u = A_{m,k} J_{m+1}(kr) - B_{m,k} Y_{m+1}(kr),$$

$$(32)$$

where $A_{m,k} = \cos(\delta_{m,k})$, $B_{m,k} = \sin(\delta_{m,k})$, and $\delta_{m,k}$

are phase shifts required to match the near and far-field solutions. The near-field ($r < r_0$) behaviour and $\delta_{m,k}$ are obtained numerically during the shooting procedure. Together, the bound and scattering states constitute a complete orthogonal eigenbasis for each $m$-sector, with the eigenvectors satisfying the inner products listed in App. G2. We can use this orthogonal eigenbasis to do perturbation theory in orders of $\alpha$, see App. G3. The resulting first order corrections to the eigenstates and eigenenergies, $\left| m, k^{(1)} \right\rangle$ and $\epsilon_{m,k}^{(1)}$, are given in Eq. (G2) and (G3). Armed with the orthogonal eigenbasis of $\sigma^z H_m$, it becomes — at least at first glance — much easier to solve Eq. (28). The idea is to expand the $a_{m,\pm\omega}$, $a_{-m,\mp\omega}^*$ fields in the bound and scattering states of the relevant $m$-sectors,

$$\begin{pmatrix} a_{m,\pm\omega}^{(1)} \\ a_{-m,\mp\omega}^{*(1)} \end{pmatrix} = \frac{c_{m,\mathrm{bd.},\pm\omega}}{\mp\omega - E_{m,\mathrm{bd.}}} \left| m, \mathrm{bd.} \right\rangle - \frac{c_{-m,\mathrm{bd.},\mp\omega}^*}{\mp\omega + E_{-m,\mathrm{bd.}}^*} \sigma^x \left| -m, \mathrm{bd.} \right\rangle^* $$
$$+ \int_0^\infty k \, dk \left( \frac{c_{m,k,\pm\omega}}{\mp\omega - E_k} \left| m, k \right\rangle - \frac{c_{-m,k,\mp\omega}^*}{\mp\omega + E_k^*} \sigma^x \left| -m, k \right\rangle^* \right),$$

$$(33)$$

where the coefficients $c_{m,\mathrm{bd.},\pm\omega}$, $c_{m,k,\pm\omega}$ are complex numbers which still need to be identified. Just like the eigenbasis in Eq. (30), we can expand the $c$'s perturbatively in $\alpha$,

$$c_{\pm m,\mathrm{bd.},\pm\omega} = \mathrm{sgn}(\gamma) c_{\pm m,\mathrm{bd.},\pm\omega}^{(0)} - i\alpha c_{\pm m,\mathrm{bd.},\pm\omega}^{(1)} + \mathcal{O}(\alpha^2)$$
$$c_{m,k,\omega} = \mathrm{sgn}(\gamma) c_{\pm m,k,\pm\omega}^{(0)} - i\alpha c_{\pm m,k,\pm\omega}^{(1)} + \mathcal{O}(\alpha^2).$$

$$(34)$$

Substituting Eq. (33) and (34) into Eq. (28), projecting $\left\langle m, \mathrm{bd.}^{(0)} \right|$ or $\left\langle m, k^{(0)} \right|$ and using the orthogonality properties given in App. 2, we find, at order $\mathcal{O}(\alpha^0)$,

$$c_{m,\mathrm{bd.},\pm\omega}^{(0)} = \left\langle m, \mathrm{bd.}^{(0)} \right| \sigma^z \left| \begin{pmatrix} f_{m,\pm\omega} \\ -f_{-m,\mp\omega}^* \end{pmatrix} \right\rangle,$$
$$c_{m,k,\pm\omega}^{(0)} = \left\langle m, k^{(0)} \right| \sigma^z \left| \begin{pmatrix} f_{m,\pm\omega} \\ -f_{-m,\mp\omega}^* \end{pmatrix} \right\rangle.$$

$$(35)$$

Following the same procedure at order $\mathcal{O}(\alpha^1)$, we calculate $c_{m,\mathrm{bd.},\pm\omega}^{(1)}$ and $c_{m,k,\pm\omega}^{(1)}$, listing the resulting expressions in Eq. (G4). Let us take a closer look at Eq. (34) and (G4) to check that these coefficients are well-defined. For $c_{m,\mathrm{bd.},\pm\omega}^{(0,1)}$, the presence of the bound state $\langle m, \mathrm{bd.}|$ ensures that the integrands are bounded, consequently the $c_{m,\mathrm{bd.},\pm\omega}^{(0,1)}$ coefficients are always well-defined. For the scattering state coefficients $c_{m,k,\pm\omega}^{(0)}$ this is no longer true, as $\left\langle m, k^{(0)} \right|$ only decays with amplitude $1/\sqrt{r}$, so that the integrand has overall $r$-dependence $\sqrt{r} f_{m,\pm\omega}(r)$. We thus need $\lim_{r \gg r_0} f_{m,\pm\omega}$ to decay faster than $1/\sqrt{r}$ for the integral to be well-defined. In Fig. 2(a), we plot $f_{m,\pm\omega}$ as a function of $r$ for $m = 0, \pm 1$, using the definitions given in

Eq. (26). While $f_0, f_1$ both decay to zero as $r \gg r_0$, $f_{-1}$ tends to a finite constant value. Physically, this comes about because $f_{-1}$ excites the ferromagnetic background rather than the skyrmion core. One way to check this is to consider what happens if we repeat our calculation for a system without a skyrmion. In this case we would have a simple ferromagnetic state with $\hat{\mathbf{M}}^{(0)} \parallel \mathbf{e}_z$ everywhere, and consequently $\theta_0(r) = 0$, from which we can confirm that the only driving component seen at linear order by the ferromagnet is $f_{-1}$. The technical conclusion from this discussion is that while we can use Eq. (34) and (G4) successfully to determine $c_{0,\pm\omega}^{(0,1)}$ in Eq. (33), we need a different approach for calculating $c_{\pm 1,\pm\omega}^{(0,1)}$. As we will show in Sec. IV.2, we can solve the problem caused by the $f_{-1}$ component by separately defining and constructing a $\left| m = -1, k = 0 \right\rangle$ mode.

## IV.2. $|m = -1, k = 0\rangle$ mode

As we are driving our system with a spatially homogeneous oscillating magnetic field, we excite the $k = 0$ magnon modes of the ferromagnetic bulk. Given that the scattering modes oscillate with wavelength $\lambda = 2\pi/k$, the $k = 0$ magnons have infinite wavelength. In other words, for $r \gg r_0$ the $|m, k = 0\rangle$ wave-functions must reach a constant $r$-independent value. To find out whether this constant value is finite or zero, we can consider the scattering states of a ferromagnet. These are given by $|m, k\rangle_u = J_{m+1}(kr)$, $|m, k\rangle_l = 0$. Using $J_n(0) = \delta_{n0}$, we conclude that $|m, 0\rangle_{u,l}$ is only finite when $m = -1$. Thus, the only relevant $k = 0$ mode in a system driven by a

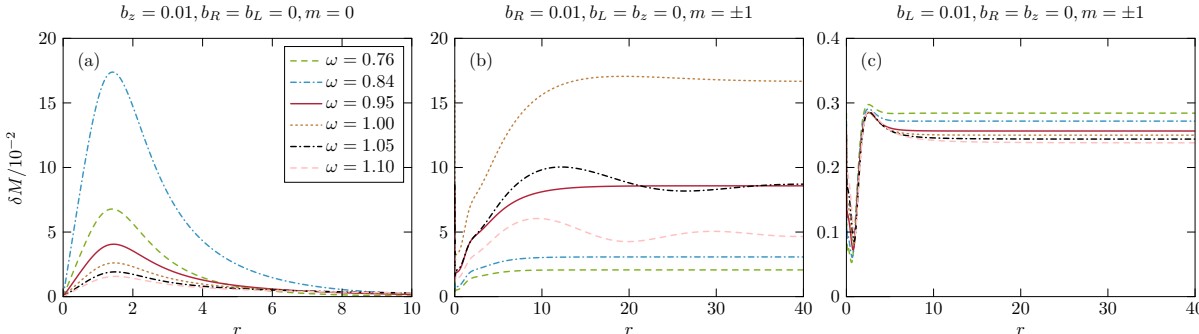

FIG. 3. $\delta M$ as a function of $r$ for different driving frequencies $\omega$ and polarisations $b_z, b_R, b_L$, with $\alpha = 0.03$, $b_0 = 1$ and $\gamma < 0$. a) Out-of-plane driving $b_z = 0.01, b_R = b_L = 0$. $\delta M$ is largest at the breathing mode resonance $\omega_{\text{br.}} = 0.839$ and is mostly confined to the region of the skyrmion, $r \lesssim r_0$. b) In-plane right-polarised driving $b_R = 0.01, b_z = b_L = 0$. The Kittel (background ferromagnet) mode at $\omega_{\text{Kittel}} = 1$ is resonantly excited. c) In-plane left-polarised driving $b_L = 0.01, b_z = b_R = 0$. The Kittel mode is *not* resonantly excited, consequently $\delta M$ is two orders of magnitude smaller than for right-polarised driving. In a material with $\gamma > 0$ the situation would be reversed, with the Kittel mode being resonantly excited by left-polarised driving. In the in-plane-polarised driving case, $\delta M$ tends to a constant value in the bulk rather than decaying to zero, due to the $k = 0$ mode. For all three driving polarisations, when $\omega > b_0 = 1$ we resonantly excite the relevant scattering mode $|m, k\rangle$ with $k = \sqrt{\omega - b_0}$, but the time-averaging washes these out in panels (a) and (c). Meanwhile, interference between the $a_{-1,+\omega}^{\text{const}}$ and $a_{-1,+\omega}^{\text{scatt}}$ gives visible oscillations in panel (b). Finite damping ensures that these oscillations decay with length scale $l \sim v_g/(\alpha\omega)$.

spatially homogeneous magnetic field is $|m = -1, k = 0\rangle$. Note that the arguments made in this paragraph are perfectly general and apply not only to a single skyrmion, but also to *any other* localised magnetic defect embedded in a ferromagnet.

While the need for a $|m = -1, k = 0\rangle$ skyrmion scattering mode is conceptually clear, trouble arises when we attempt to implement it numerically using the shooting method. The reason is the following: for the scattering modes, as long as $k$ is not extremely small, the oscillations of $|m = -1, k\rangle_u$ conveniently penetrate into the bulk of the ferromagnet without significant numerical errors. Indeed, in this case, the energy term in the eigenvalue equation starts to dominate the potentials (evaluated at machine precision) at a *finite* distance from the skyrmion core. However, as $k \to 0$, we need to go further and further away from the skyrmion core for the energy term to completely overshadow the potentials and exhibit the oscillatory nature of the wavefunction in the bulk. Unfortunately, this very fact renders the shooting numerics invalid at tiny values of $k$, as small inaccuracies in the numerical value of the potentials cannot be eliminated up to very large distances and they end up fundamentally changing the nature of the numerical solution. The manifestly unstable nature of the shooting numerics in the small $k$ regime can actually be attributed to the fact that the $|m = -1, k = 0\rangle$ mode with $\epsilon = b_0$ is a boundary mode across which the long-distance asymptotics of the (upper) wavefunction changes from exponential (for $0 < \epsilon < b_0$) to decaying oscillatory (for $\epsilon > b_0$). In fact, precisely at $k = 0$, the eigenvalue equation Eq. (29) admits two allowed asymptotic solutions, $\lim_{r \gg r_0} |m = -1, k = 0\rangle_u = A + B \log(r)$. Physical intuition then dictates that $B = 0$, but numerical inaccuracies in the code (which are unavoidable) always result

in a finite $B$. An alternative approach to the shooting method is therefore needed. Here we will propose a solution where we write the $|m = -1, k = 0, \pm\omega\rangle$ mode as a superposition of $f_{\pm1, \pm\omega}$ and all the other modes from the $m = -1$ sector,

$$|-1, 0, \pm\omega\rangle = (\text{sgn}(\gamma) + i\alpha\sigma^z)^{-1} \begin{pmatrix} f_{-1,\pm\omega} \\ -f_{1,\mp\omega}^* \end{pmatrix}$$
$$+ \int_{k>0}^\infty k\,dk\left(\tilde{c}_{-1,k,\pm\omega} |-1, k\rangle - \tilde{c}_{1,k,\mp\omega}^* \sigma^x |1, k\rangle^*\right)$$
$$+ \tilde{c}_{-1,\text{trans.},\pm\omega} |-1, \text{trans.}\rangle, \qquad (36)$$

where $f_{-1,\pm\omega}$ takes care of the upper component tending to a constant value as $r \gg r_0$. The role of all the other $m = -1$ modes is to ensure that the eigenvalue equation,

$$E_0(\text{sgn}(\gamma) + i\alpha\sigma^z) |-1, 0, \pm\omega\rangle = \sigma^z H_{-1} |-1, 0, \pm\omega\rangle, \qquad (37)$$

is fulfilled. Perturbatively expanding the $\tilde{c}_{m,k,\pm\omega}^{(0,1)}$'s analogously to the $c_{m,k,\pm\omega}^{(0,1)}$'s in Eq. (34), we find

$$\tilde{c}_{-1,k,\pm\omega}^{(0)} = -\frac{1}{k^2}\left\langle -1, k^{(0)}\middle| (H_{-1} - b_0\sigma^z)\middle| \begin{pmatrix} f_{-1,\pm\omega} \\ -f_{1,\mp\omega}^* \end{pmatrix} \right\rangle, \qquad (38)$$

$$\tilde{c}_{1,k,\mp\omega}^{*(0)} = \frac{1}{2b_0 + k^2}\left\langle 1, k^{(0)}\middle| \sigma^x(H_{-1} - b_0\sigma^z)\middle| \begin{pmatrix} f_{-1,\pm\omega} \\ -f_{1,\mp\omega}^* \end{pmatrix} \right\rangle$$

and $\tilde{c}_{-1,\text{trans.},\pm\omega}^{(0,1)} = 0$, with the $\tilde{c}_{\pm1,k,\pm\omega}^{(1)}$'s given in Eq. (G6). To check that the integrands in all the $\tilde{c}^{(0,1)}$'s are now bounded, let us look at the $r \gg r_0$ limit of the $H_{-1} - b_0\sigma^z$ and $(b_0 - H_{-1})\sigma^z$ operators found inside these expressions. Using $\lim_{r \gg r_0} H_{-1} = \mathbb{1}b_0 + \frac{2}{r^2}(\mathbb{1} - \sigma^z)$ we

have

$$\lim_{r \gg r_0} (H_{-1} - b_0 \sigma^z) = (\mathbb{1} - \sigma^z)\left(b_0 + \frac{2}{r^2}\right),$$

$$\lim_{r \gg r_0} (b_0 - H_{-1})\sigma^z = -\frac{2}{r^2}(\mathbb{1} - \sigma^z).$$

The presence of the $(\mathbb{1} - \sigma^z)$ factor in both of these terms guarantees that the $f_{-1,\pm\omega}$ component, which was caus-

ing the problematic unbounded behaviour in the integrands of Eq. (35) and (G4), gets pushed out of the integrand at large $r$. This makes the integrands overall bounded, as they vanish for $r \gtrsim r_0$. Consequently, the $\tilde{c}^{(0,1)}_{\pm 1,\pm\omega}$'s coefficients are completely well-defined.

We can use the newly defined $|-1, 0, \pm\omega\rangle$ mode from Eq. (36) to write an ansatz for the $a_{1,\omega}$, $a^*_{-1,\omega}$ fields,

$$\begin{pmatrix} a^{(1)}_{1,\pm\omega} \\ a^{*(1)}_{-1,\mp\omega} \end{pmatrix} = -\frac{1}{\mp\omega + E_0^*}\sigma^x |-1, 0, \mp\omega\rangle^* - \int_{k>0}^\infty k\,dk \left(\frac{\tilde{c}_{1,k,\pm\omega}}{\mp\omega - E_k}|1, k\rangle - \frac{\tilde{c}^*_{-1,k,\mp\omega}}{\mp\omega + E_k^*}\sigma^x |-1, k\rangle^*\right). \qquad (39)$$

Using the eigenvalue equation Eq. (29), it is straightforward to verify that Eq. (39) solves Eq. (28). The technique which we used here of separating the $k = 0$ mode from all the finite $k$ modes is inspired by the well-known trick of separating the BEC mode from all the finite energy excitations in a bosonic gas [26]. Here, however, the defining feature of the $k = 0$ mode is not that it costs zero energy (it costs finite energy $b_0$) — rather, its "specialness" is observed in the first order dynamics of the driven system. Inspecting Eq. (39), we can see that when driven on-resonance, only the $k = 0$ mode will be resonantly excited and grow like $1/\alpha$ in the limit of small damping. On the other hand, any resonance excited in the finite $k$-modes subsequently gets smeared out by the integration over $k$.

In Fig. 2(d) we plot $c^{(0,1)}_{0,k,\pm\omega}$, $\tilde{c}^{(0,1)}_{\pm 1,k,\pm\omega}$ as a function of $k$. All the coefficients decay to zero for $k \gtrsim 6$, which provides a natural upper $k$-cutoff for the numerical implementation of expressions Eq. (33) and (39). Note the presence of the $1/k^2$ factor in $\tilde{c}^{(0,1)}_{-1,k,\mp\omega}$, which causes a singularity as $k \to 0$. While this would be problematic if we really wanted to obtain the $k = 0$ mode in isolation, when combined with the finite $k$ scattering modes in Eq. (39) the $1/k^2$ singularities vanish, because for each $\tilde{c}^{(0,1)}_{-1,k,\mp\omega}$ there is a pre-factor $1/(\pm\omega + E_0^*) - 1/(\pm\omega + E_k^*) \sim k^2$. As we have full numerical knowledge of the $c_{0,k,\pm\omega}$, $\tilde{c}_{\pm 1,k,\pm\omega}$ coefficients, we can calculate the linear response fields $a^{(1)}_{0,\pm\omega}$, $a^{(1)}_{\pm 1,\pm\omega}$ for any $r$ using Eq. (33) and (39).

We now consider the far-field limit $r \gg r_0$, where only the constant component of the $k = 0$ mode and the finite $k$ scattering mode contributions survive. We denote these as $a^{(1),\text{const}}_{m,\pm\omega}$ and $a^{(1),\text{scatt}}_{m,\pm\omega}$, respectively. $a^{(1),\text{const}}_{-1,\pm\omega}$ can be directly read off from Eq. (36) and (39),

$$a^{(1),\text{const}}_{-1,\pm\omega} = -\frac{e^{-ih}b_{R/L}}{2\sqrt{2}}\frac{1}{\pm\omega + \text{sgn}(\gamma)b_0 - ib_0\alpha}, \qquad (40)$$

while $a^{(1),\text{const}}_{1,\pm\omega}$, $a^{(1),\text{const}}_{0,\pm\omega} = 0$ as there is no finite $k = 0$ mode in those $m$-sectors. As the scattering modes adopt the free Bessel form in Eq. (32), $a^{(1),\text{scatt}}_{m,\pm\omega}$ can be further

simplified using a contour integral, see App. I for technical details. We then obtain, for $\gamma < 0$ and to leading order in $\alpha$,

$$a^{(1),\text{scatt}}_{0,+\omega} = -\sqrt{\frac{\pi}{2k_0 r}}c^{(0)}_{0,k_0,+\omega}e^{-i\delta_{0,k_0}}e^{i(\frac{\pi}{4}-k_0 r)}e^{-\alpha\omega r/v_g},$$

$$(41)$$

$$a^{(1),\text{scatt}}_{\pm 1,+\omega} = \sqrt{\frac{\pi}{2k_0 r}}\tilde{c}^{(0)}_{\pm 1,k_0,+\omega}e^{i(\pm\frac{\pi}{2}-\delta_{\pm 1,k_0})}e^{i(\frac{\pi}{4}-k_0 r)}e^{-\alpha\omega r/v_g},$$

and $a^{(1),\text{scatt}}_{0,-\omega} = a^{(1),\text{scatt}}_{\pm 1,-\omega} = 0$. Here $k_0 = \sqrt{\omega - b_0}$ is the momentum of the resonantly selected spin-wave and $v_g = 2\sqrt{\omega - b_0}$ is its group velocity. The physical interpretation is that when driven with $\omega > b_0$, the skyrmion acts like a resonant antenna, sending out magnons with momentum $k_0$ which get damped over a length scale $l \sim v_g/(\alpha\omega)$.

Videos of the time-dependent linear response of the skyrmion to out-of-plane as well as in-plane right-polarised driving are provided as supplementary materials, see App. A. Another way to visualise the linear response of the system is to consider the time- and polar angle- averaged local deviation from the static skyrmion texture $\mathbf{M}^{(0)}$, defined as $\langle|\delta\hat{\mathbf{M}}|^2\rangle = \langle|\hat{\mathbf{M}}(t) - \hat{\mathbf{M}}^{(0)}|^2\rangle_{\chi,t} = 2\langle a^* a\rangle_{\chi,t}$, using Eq. (17). To leading order $\mathcal{O}(\epsilon^2)$, this is given by

$$\langle|\delta\hat{\mathbf{M}}|^2\rangle = \sum_{\substack{m=0,\pm 1 \\ \omega'=\pm\omega}} 2a^{*(1)}_{m,\omega'}(r)a^{(1)}_{m,\omega'}(r). \qquad (42)$$

In Fig. 3 we plot $\delta M = \sqrt{\langle|\delta\hat{\mathbf{M}}|^2\rangle}$ as a function of $r$ for various driving frequencies and polarisations with $\gamma < 0$ and $\alpha = 0.03$. For out-of-plane driving, $b_L = b_R = 0$, $\delta M$ is mostly confined to the skyrmion radius and largest when we drive at the breathing mode frequency, $\omega = \omega_{\text{br.}} = 0.839$, see Fig. 3(a). For in-plane driving, $b_z = 0$, the resonant frequency is the Kittel frequency, $\omega_{\text{Kit.}} = b_0 = 1$, but it is only resonantly excited by right-polarised driving, as can be seen by comparing the scales in Fig. 3(b) and (c). In-plane driving excites the ferromagnetic bulk, hence $\delta M$ tends to a constant value in the

ferromagnetic bulk rather than decaying to zero, as was the case with out-of-plane driving. Interference between the $a_{-1,+\omega}^{(1),\text{const}}$ and $a_{-1,+\omega}^{(1),\text{scatt}}$ results in visible oscillations of wavelength $2\pi/k_0$ in Fig. 3(b) when $\omega > \omega_{\text{Kit.}}$.

In this section we developed all the machinery necessary to obtain the full linear response of the skyrmion to external driving by a homogeneous magnetic field $\mathbf{B}_1(t)$. Next, we will use this to calculate the second order translational velocity $\mathbf{v}_{\text{trans}}$ in Sec. V.

## V. SWIMMING SKYRMION

In the linear response framework of Sec. IV, the skyrmion was able to undergo periodic contractions, dilations and rotational-symmetry breaking deformations, but its (time-averaged) centre remained firmly stuck in one place. At order $\mathcal{O}(\epsilon^2)$, however, the skyrmion will start to "swim" with constant velocity $\mathbf{v}_{\text{trans}}$, see also the video provided in App. A. This motion happens as a *consequence* of the linear order cyclic asymmetric deformations, a kind of mechanism also used by jellyfish when they swim through water. In this section we will calculate $\mathbf{v}_{\text{trans}}$ and understand how it is influenced by the driving field's polarisation and frequency, as well as the amount of damping in the system.

We wish to solve Eq. (12) for $\mathbf{v}_{\text{trans}}$. On the LHS of this equation we have the familiar gyrocoupling and dissipation terms of the Thiele equation, which for a single skyrmion are given by

$$
\begin{aligned}
\tilde{\mathbf{G}} &= -4\pi\mathbf{e}_z, \\
\tilde{\mathcal{D}}_{xx} = \tilde{\mathcal{D}}_{yy} &= \pi \int_0^\infty dr\, r \left( \theta_0'^2 + \frac{1}{r^2}\sin^2(\theta_0) \right),
\end{aligned}
\tag{43}
$$

where we used the skyrmion's translational symmetry in the $\mathbf{e}_z$-direction to define $\tilde{G} = G/L_z$, $\tilde{\mathcal{D}} = \mathcal{D}/L_z$, $L_z$ being the sample length in the $\mathbf{e}_z$-direction. Due to the skyrmion's rotational symmetry, $\tilde{\mathcal{D}}_{xx}$ and $\tilde{\mathcal{D}}_{yy}$ are the only non-zero entries of the dissipation matrix $\tilde{\mathcal{D}}_{ij}$. As a result, the $v_{\text{trans}}^z$ component vanishes completely and Eq. (12) reduces to a $2 \times 2$ matrix equation,

$$
\begin{pmatrix} \alpha\tilde{\mathcal{D}}_{xx} & -4\pi\text{sgn}(\gamma) \\ 4\pi\text{sgn}(\gamma) & \alpha\tilde{\mathcal{D}}_{yy} \end{pmatrix} \begin{pmatrix} v_{\text{trans}}^x \\ v_{\text{trans}}^y \end{pmatrix} = \begin{pmatrix} F_{\text{trans}}^x/L_z \\ F_{\text{trans}}^y/L_z \end{pmatrix},
\tag{44}
$$

which we can easily invert to evaluate $\mathbf{v}_{\text{trans}} = v_{\text{trans}}^x\mathbf{e}_x + v_{\text{trans}}^y\mathbf{e}_y$. As discussed in Sec. III, there are two different ways of evaluating the force on the RHS of Eq. (44), given in Eq. (14) and (16). In both methods, the force densities are functions of $\hat{\mathbf{M}}^{(0)}$, $\mathbf{M}^{(1)}$, which carry no $z$-dependence. This means we can replace $\frac{1}{L_z}\int d^3r \rightarrow \int d^2r$ in Eq. (14) and (16).

Let us first develop some intuition for $\mathbf{F}_{\text{trans}}$ using the first method, Eq. (14). It turns out that the first term in Eq. (14) actually vanishes in the case of a driven

skyrmion,

$$
\int d^2r\, \langle \hat{\mathbf{M}}^{(0)} \cdot (\dot{\mathbf{M}}^{(1)} \times \nabla_i\mathbf{M}^{(1)})\rangle_t = 0.
\tag{45}
$$

This result relies on $\alpha$ being finite so that all the linear response, with the exception of the spatially constant bulk contribution coming from the $k = 0$ mode, decays to zero as $r \to \infty$, see App. J for details. For this reason, Eq. (45) is in fact a general result, valid for any driven magnetic system with some localised topological charge embedded in a ferromagnetic sea, as long as there is finite damping. Physically, Eq. (45) has two interesting consequences. The first involves the emergent electric field, which is given by

$$
E_i = \frac{\hbar}{2|e|}\hat{\mathbf{M}} \cdot (\nabla_i\hat{\mathbf{M}} \times \dot{\hat{\mathbf{M}}})
\tag{46}
$$

for any general moving magnetic texture $\mathbf{M}$ [27]. To order $\mathcal{O}(\epsilon^2)$, Eq. (46) has two contributions, $\frac{\hbar}{2|e|}\hat{\mathbf{M}}^{(0)} \cdot (\nabla_i\mathbf{M}^{(1)} \times \dot{\mathbf{M}}^{(1)})$ and

$$
\frac{\hbar}{2|e|}\hat{\mathbf{M}}^{(0)} \cdot (\nabla_i\mathbf{M}^{(0)} \times \dot{\mathbf{M}}^{(2)}) = \frac{\hbar}{2|e|}(\mathbf{q}^{\text{top}} \times \mathbf{v}_{\text{trans}})_i,
$$

where $\mathbf{q}^{\text{top}}$ is the topological charge density defined in Eq. (C8). When time-averaged and integrated over all space, the first contribution vanishes and we conclude that the $\mathcal{O}(\epsilon^2)$ time-averaged, spatially integrated electric field is just given by

$$
\int d^2r\, \langle \mathbf{E}^{(2)}\rangle_t = \frac{\hbar}{2|e|}\mathbf{G} \times \mathbf{v}_{\text{trans}}.
\tag{47}
$$

This means that an electron far away will only feel an electric field from the *net time-averaged* motion of the skyrmion at velocity $\mathbf{v}_{\text{trans}}$. Eq. (47) is exactly the electric field predicted by Faraday's law of induction $\dot{\mathbf{B}} = -\boldsymbol{\nabla} \times \mathbf{E}$, if we model the moving skyrmion as a point flux with an emergent magnetic field, $\mathbf{B} = -\frac{h}{|e|}\delta(\mathbf{r} - \mathbf{v}_{\text{trans}}t)\mathbf{e}_z$.

The second consequence of Eq. (45) is that the time-averaged total rate of change of the magnon momentum, $\int d^2r\langle\dot{P}_i^{\text{m}}\rangle$, defined in Eq. (C9), also vanishes. The physical interpretation of this is that all the force generated by the time-averaged rate of change of momentum of the emitted magnons eventually gets absorbed by the bulk due to the finite damping $\alpha$.

Getting back to our Thiele equation, we only have the second force term to worry about in Eq. (44),

$$
\begin{aligned}
F_{\text{trans}}^i &= \alpha \int d^2r\langle \dot{\mathbf{M}}^{(1)} \cdot \nabla_i\mathbf{M}^{(1)}\rangle_t \\
&= \alpha \int d^2r\Big\langle \dot{a}^{(1)}\nabla_i a^{*(1)} + \dot{a}^{*(1)}\nabla_i a^{(1)} \\
&\quad - i\cos(\theta_0)\nabla_i(\chi)(\dot{a}^{(1)}a^{*(1)} - \dot{a}^{*(1)}a^{(1)})\Big\rangle_t,
\end{aligned}
\tag{48}
$$

where we rescaled $F_{\mathrm{trans}}^i/L_z \to F_{\mathrm{trans}}^i$ to avoid extra clutter, and then substituted the expansion Eq. (17) to write the integrand in terms of the $a^{(1)}$, $a^{*(1)}$ fields. Due to the spatial derivative $\nabla_i$ and integration over the polar angle $\chi$, only terms carrying a net angular momentum $m = \pm 1$ in the integrand of Eq. (48) will survive. For example, terms such as $a_{0,+\omega}^{(1)} a_{1,+\omega}^{*(1)}$, $a_{0,+\omega}^{(1)} a_{-1,+\omega}^{*(1)}$ survive but $a_{0,+\omega}^{(1)} a_{0,+\omega}^{*(1)}$, $a_{-1,+\omega}^{(1)} a_{-1,-\omega}^{*(1)}$ are killed by the integration over $\chi$. All surviving terms in the integrand of Eq. (48) are therefore products of one $a_0^{(1)}$ field and one $a_{\pm 1}^{(1)}$ field — we thus require a *tilted* driving field $\mathbf{b}_1(t)$, with $b_z > 0$ and at least one of $b_R, b_L > 0$, to get a finite $\mathbf{F}_{\mathrm{trans}}$.

Next, we consider what happens to $\mathbf{F}_{\mathrm{trans}}$ as function of the driving frequency $\omega$ and damping $\alpha$. If we drive below the gap, $\omega < \omega_{\mathrm{Kit.}}$, none of the scattering modes are resonantly excited, so $a_{0,\pm\omega}^{(1)}$, and consequently the integrand of $\mathbf{F}_{\mathrm{trans}}$, are bounded to the skyrmion radius $r_0$. We choose to discuss in particular the case of a Néel skyrmion, $h = 0$, driven with a general tilted field $\mathbf{b}_1(t)$ as defined in Eq. (2), with $b_z, b_R, b_L > 0$ and phase shift $\delta = 0$. A different choice of $\delta$ or $h$ would have an effect on the orientation, but not the magnitude of $\mathbf{F}_{\mathrm{trans}}$. With this particular choice, the forces off and on resonance in the limit of small damping are given by

$$\lim_{\alpha \to 0} \mathbf{F}_{\mathrm{trans}}(\omega \neq \omega_{\mathrm{br.}}, \omega_{\mathrm{Kit.}}) \sim \alpha \mathbf{e}_y,$$
$$\lim_{\alpha \to 0} \mathbf{F}_{\mathrm{trans}}(\omega = \omega_{\mathrm{br.}}, \omega_{\mathrm{Kit.}}) \sim \mathbf{e}_x,$$

(49)

respectively, see App. K for the derivation of this result. In the resonant driving cases, $\omega = \omega_{\mathrm{br.}}, \omega_{\mathrm{Kit.}}$, the prefactor $\alpha$ gets cancelled by a $1/i\alpha$ singularity entering via either the $a_{0,\pm\omega}^{(1)}$ or $a_{-1,\pm\omega}^{(1)}$ fields, depending on whether we are at the breathing or Kittel resonance. Meanwhile, on the other side of the Thiele equation only the gyrocoupling term survives in the limit of small $\alpha$, resulting in velocities

$$\lim_{\alpha \to 0} \mathbf{v}_{\mathrm{trans}}(\omega \neq \omega_{\mathrm{br.}}, \omega_{\mathrm{Kit.}}) \sim \alpha \mathbf{e}_x,$$
$$\lim_{\alpha \to 0} \mathbf{v}_{\mathrm{trans}}(\omega = \omega_{\mathrm{br.}}, \omega_{\mathrm{Kit.}}) \sim \mathbf{e}_y,$$

(50)

off and on resonance. We conclude that when driven non-resonantly below the gap, the skyrmion curiously "swims" faster in more damped environments! As damping needs to be non-zero for the skyrmion to move in this driving frequency range, we call it the *friction-driven* regime. In other parts of the natural world, snails, caterpillars and other molluscs are also known to rely on similar friction-driven mechanisms for their motion.

If we instead drive above the gap, $\omega > \omega_{\mathrm{Kit.}}$, the skyrmion turns into a magnon antenna, emitting magnons of radial momentum $k_0 = \sqrt{\omega - \omega_{\mathrm{Kit.}}}$ and angular momentum $m = 0, \pm 1$ in a non-uniform way. This means that the integrand of $\mathbf{F}_{\mathrm{trans}}$ is no longer bounded to $r_0$, but remains finite up to the magnon's decay length, $l = v_g/(\alpha\omega)$. Thus, for small $\alpha$ most of the integrand is

actually *outside* the skyrmion radius $r_0$. In the region $r \gg r_0, 2\pi/k_0$ we can substitute the far-field expressions $a_{0,\pm 1}^{(1),\mathrm{const}}$ and $a_{0,\pm 1}^{(1),\mathrm{scatt}}$, given in Eq. (40) and (41), to evaluate the integrand. The slowest decaying terms in the integrand come from products such as $a_{0,+\omega}^{(1),\mathrm{scatt}} a_{1,+\omega}^{*(1),\mathrm{scatt}}$, and have radial dependence $\sim e^{-2\alpha\omega r/v_g}/r$. From dimensional analysis, we know that the radial integral of this quantity scales as

$$\int_{r_0}^{\infty} \frac{r\, dr\, e^{-2\alpha\omega r/v_g}}{r} \sim \frac{1}{\alpha}.$$

The $\alpha$ pre-factor in Eq. (48) will cancel the $1/\alpha$ singularity coming from the integral and we conclude that

$$\lim_{\alpha \to 0} F_{\mathrm{trans}}(\omega > \omega_{\mathrm{Kit.}}) \sim \mathrm{const.} \qquad (51)$$

In the limit of small damping, the force is therefore *independent* of $\alpha$ when we drive above the ferromagnetic gap. On the other side of the Thiele equation the gyrocoupling term still dominates for small $\alpha$, so $\lim_{\alpha \to 0} v_{\mathrm{trans}}(\omega > \omega_{\mathrm{Kit.}}) \sim \mathrm{const.}$

Collecting all these results in one expression for the speed of the skyrmion as a function of the driving frequency $\omega$, we have

$$\lim_{\alpha \to 0} v_{\mathrm{trans}} \sim \begin{cases} \alpha, \ \omega < \omega_{\mathrm{Kit.}}, \omega \neq \omega_{\mathrm{br.}}, \\ \mathrm{const.}, \ \omega = \omega_{\mathrm{br.}} \ \mathrm{or}\ \omega \geq \omega_{\mathrm{Kit.}}. \end{cases} \qquad (52)$$

The fact that the skyrmion's speed is independent of $\alpha$ when we drive it above the ferromagnetic gap suggests that it cannot be the same friction-driven mechanism that is causing the skyrmion's motion in this frequency range. We can gain a better understanding of the mechanism by looking at the $\mathcal{O}(\epsilon^2)$ contributions of the time-averaged magnon rate-of-change-of-momentum and current densities, $\langle \dot{P}_\nu^{\mathrm{m}} \rangle_t$ and $\langle \mathbf{J}_\nu^{(2)} \rangle_t$, which we defined in Eq. (C5) and (C9). These quantities enter the total force density $\mathfrak{F}^{\mathrm{tot}}$ in Eq. (C10), but disappear from the equation once we integrate Eq. (C10) over all space. In fact we can also integrate Eq. (C10) over a disk of finite radius, rather than all space. We expect that at some finite radius $r^{\mathrm{bulk}}(\omega)$ the time-averaged magnon rate-of-change-of-momentum and current densities will reach the ferromagnetic bulk values

$$\langle \dot{P}_\nu^{\mathrm{m}} \rangle_t^{\mathrm{bulk}} = 0,$$
$$\langle \mathbf{J}_\nu^{(2)} \rangle_t^{\mathrm{bulk}} = \mathrm{const.}\, \mathbf{e}_\nu.$$

Once they have reached these bulk values, $\langle \dot{P}_\nu^{\mathrm{m}} \rangle_t$ and $\nabla \cdot \langle \mathbf{J}_\nu^{(2)} \rangle$ will once again vanish after the integration step, so it is in fact sufficient to integrate Eq. (C10) in the region $0 < r < r^{\mathrm{bulk}}(\omega)$. $r^{\mathrm{bulk}}(\omega)$ depends strongly on the driving frequency $\omega$. In Fig. 5 we plot the driven skyrmion spin texture together with $\langle \mathbf{J}_\nu^{(2)} \rangle_t$ and $\langle \dot{P}_\nu^{\mathrm{m}} \rangle_t$ for $\nu = x, y$, at three different driving frequencies,

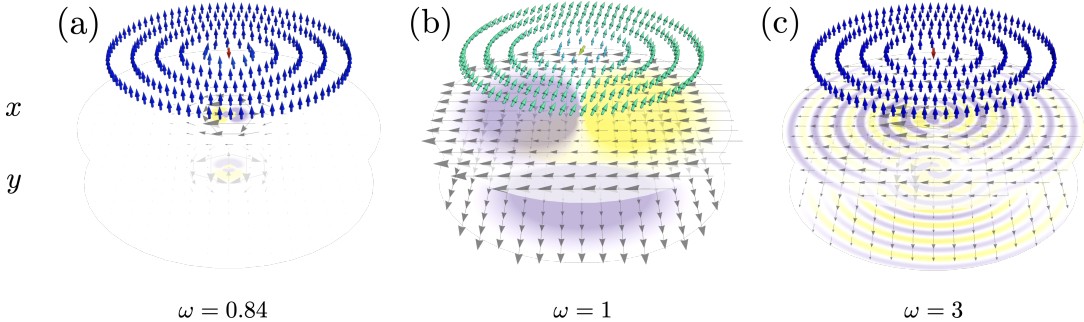

$\omega = 0.84$          $\omega = 1$          $\omega = 3$

FIG. 4. A Néel skyrmion driven (a) below the gap, $\omega = 0.84$, (b) at the gap, $\omega = 1$, and (c) above the gap, $\omega = 3$. All other parameters are fixed, with $b_z = b_R = 0.03$, $b_L = 0$, $\alpha = 0.01$ and $\delta = h = 0$. In each panel, the top layer shows the skyrmion spin texture. The next two layers show the time-averaged $\mathcal{O}(\epsilon^2)$ magnon rate-of-change-of-momentum and current densities $\langle \dot{P}_\nu^{\mathrm{m}} \rangle_t$ and $\langle \mathbf{J}_\nu^{(2)} \rangle_t$, defined in Eq. (C5) and (C9), for $\nu = x, y$, respectively. Purple and yellow indicate the minimum and maximum values of $\langle \dot{P}_\nu^{\mathrm{m}} \rangle_t$, respectively, and white corresponds to $\langle \dot{P}_\nu^{\mathrm{m}} \rangle_t = 0$. The arrows indicate the size and direction of the time-averaged local magnon current $\langle \mathbf{J}_\nu^{(2)} \rangle_t$, which for $r \gtrsim r^{\mathrm{bulk}}(\omega)$ tends to $\langle \mathbf{J}_\nu^{(2)} \rangle_t^{\mathrm{bulk}} = -\frac{1}{2} b_0 \langle \mathbf{M}^{(1)} \cdot \mathbf{M}^{(1)} \rangle_t \mathbf{e}_\nu$. In panel (a), $\langle \dot{P}_\nu^{\mathrm{m}} \rangle_t$ and $\langle \mathbf{J}_\nu^{(2)} \rangle_t$ reach their bulk values already at $r \gtrsim r_0$. This is not the case when $\omega \geq \omega_{\mathrm{Kit.}}$, panels (b) and (c). In panel (b), we are resonantly driving the $k = 0$ mode. This resonance manifests as a very obvious decrease in the time-averaged $z$-component of the ferromagnetic spins, $\langle \hat{\mathbf{M}}_z \rangle_t$, and is signalled by the lighter green colour of the bulk spins in panel (b). In panel (c), we excite a scattering state magnon of frequency $k_0 = \sqrt{\omega - \omega_{\mathrm{Kit.}}}$, whose decay length $l \sim v_g/(\alpha \omega) \gg r_0$. We can conclude that in the vicinity of the skyrmion $r \gtrsim r_0$, the magnon density and current make no contribution to the total force density $\mathfrak{F}_\nu^{\mathrm{tot}}$ (defined in Eq. (C10)) when $\omega < \omega_{\mathrm{Kit.}}$, but do contribute significantly when $\omega > \omega_{\mathrm{Kit.}}$.

$\omega = 0.84, 1$ and 3. In Fig. 5(a), where $\omega < \omega_{\mathrm{Kit.}}$, we see that $\langle \dot{P}_\nu^{\mathrm{m}} \rangle_t$ and $\langle \mathbf{J}_\nu \rangle_t$ already reach their bulk values at $r \sim r_0$, as no finite $k$ magnons are excited below the gap. On the other hand, in Fig. 5(c), where $\omega > \omega_{\mathrm{Kit.}}$, magnons of finite $k_0 = \sqrt{\omega - \omega_{\mathrm{Kit.}}}$ get excited. As a consequence, $\langle \dot{P}_\nu^{\mathrm{m}} \rangle_t$ and $\langle \mathbf{J}_\nu \rangle_t$ do not reach their bulk values until all these magnons have decayed, which happens at $r \sim l = v_g/(\alpha \omega)$. Fig. 5(b) is a bit special, as there the $k = 0$ mode is resonantly excited. This $k = 0$ magnon formally has a decay length $l = 0$, as its group velocity $v_g = 0$, but we can see nevertheless that $r^{\mathrm{bulk}}(\omega) > r_0$. At $\omega = \omega_{\mathrm{Kit.}}$, the skyrmion is maximally relying on the resonant excitation of the ferromagnetic background to move. Certain kinds of "lazy" jellyfish analogously rely on ocean currents to help them move, instead of wasting precious energy propelling themselves on their own. We summarise the frequency dependence of $r^{\mathrm{bulk}}(\omega)$ in the following expression,

$$r^{\mathrm{bulk}}(\omega) \sim \begin{cases} r_0, \omega < \omega_{\mathrm{Kit.}}, \\ l = v_g/(\alpha \omega), \omega > \omega_{\mathrm{Kit.}}. \end{cases} \quad (53)$$

In the limit of small damping, $r^{\mathrm{bulk}}(\omega)$ therefore experiences a large jump as $\omega$ crosses the ferromagnetic gap.

It is also perfectly legal (although more complicated, due to non-vanishing boundary terms), to integrate Eq. (C10) between $r_{\mathrm{min}} = 0$ and $r_0 < r_{\mathrm{max}} < r^{\mathrm{bulk}}(\omega)$. Below the gap, $\omega < \omega_{\mathrm{Kit.}}$, this change makes no difference to which of the terms in Eq. (C10) dominate. $\langle \dot{P}_\nu^{\mathrm{m}} \rangle_t$ and $\langle \mathbf{J}_\nu^{(2)} \rangle_t$ will already have reached their bulk values by $r = r_0$, so they will vanish after integration, leaving the friction term $\alpha \dot{\mathbf{M}} \cdot \nabla_\nu \mathbf{M}$ as the sole contributor to

$\mathbf{F}_{\mathrm{trans}}$. However, if $\omega > \omega_{\mathrm{Kit.}}$, $\langle \dot{P}_\nu^{\mathrm{m}} \rangle_t$ and $\langle \mathbf{J}_\nu^{(2)} \rangle_t$ will not yet have reached their bulk values at $r = r_{\mathrm{max}}$, and they will dominate the friction term $\alpha \dot{\mathbf{M}} \cdot \nabla_\nu \mathbf{M}$ if $\alpha$ is very small. In this case, $\mathbf{F}_{\mathrm{trans}}$ — as calculated in the vicinity of the skyrmion — mostly originates from emitted magnons, rather than friction. We can therefore say that in the $\omega > \omega_{\mathrm{Kit.}}$ range, the skyrmion moves by a *magnon emission* mechanism, in which asymmetrically emitted magnons result in a momentum counter kick to the skyrmion, which causes it to move in the opposite direction to ensure the overall conservation of linear momentum in the system. We should however keep in mind that if we extend $r_{\mathrm{max}}$ to $r^{\mathrm{bulk}}(\omega)$ and above, all magnon momentum eventually gets lost through damping to the surroundings, leaving only the $\alpha \dot{\mathbf{M}} \cdot \nabla_\nu \mathbf{M}$ friction contribution in $\mathbf{F}_{\mathrm{trans}}$.

We now wish to obtain some concrete numerical values for $\mathbf{F}_{\mathrm{trans}}$ and the resulting $\mathbf{v}_{\mathrm{trans}}$. If we were to use Eq. (14) for this purpose, we would have to calculate all the $a^{(1)}(r)$, $a^{*(1)}(r)$ fields up to $r = r^{\mathrm{bulk}}(\omega)$, which can get very large at small $\alpha$ for $\omega > \omega_{\mathrm{Kit.}}$, see Eq. (53). In principle, the far-field analytical expressions $a^{(1),\mathrm{scatt}}$, $a^{(1),\mathrm{const}}$ eventually become good approximations to the numerically evaluated $a^{(1)}$, $a^{*(1)}$, but this only happens after $r \geq 2\pi/k_0$. The low $k$ scattering modes, excited when $\omega \gtrsim \omega_{\mathrm{Kit.}}$, therefore continue to pose a numerical challenge, as the analytical approximations of $a^{(1)}(r), a^{*(1)}(r)$ for these modes only starts to be accurate at very large $r$. We can avoid these headaches if we instead use Eq. (16) to calculate $\mathbf{F}_{\mathrm{trans}}$. The great advantage of Eq. (16) is that the integrand is always bounded to the skyrmion radius, because the

terms $\nabla_i \hat{\mathbf{M}}^{(0)}$ and $\nabla_i(\hat{\mathbf{M}}^{(0)} \cdot \mathbf{B}_{\text{eff}}^{(0)})$ both vanish for $r \gtrsim r_0$. Importantly, this happens *independently* of the driving frequency $\omega$, as $\hat{\mathbf{M}}^{(0)}$ and $\mathbf{B}_{\text{eff}}^{(0)}$ describe only the static texture and don't know anything about the driving. Of course, there is a price to pay for these numerical advantages — once evaluated in terms of $a^{(1)}, a^{*(1)}$, Eq. (16) is algebraically much uglier and harder to interpret than Eq. (14), see Eq. (H1) for the full expression. Nevertheless with some help from *Mathematica* these algebraic difficulties disappear and we can comfortably use Eq. (H1) for the numerical evaluation of $\mathbf{F}_{\text{trans}}$.

Let us now choose some parameters for the drive, while also trying to keep it as general as possible. We need a tilted driving field $\mathbf{b}_1(t)$ to get a finite $v_{\text{trans}}$, so $b_z$ must in any case be non-zero. For the in-plane driving field, we have two degrees of freedom in the choice of $b_R$ and $b_L$. As Eq. (48) does not contains any cross-terms $\propto b_R b_L$, it is sufficient to consider the cases $b_R = 0$ and $b_L = 0$ separately to obtain the full $\mathbf{v}_{\text{trans}}$. Any mixed in-plane driving with both $b_R$ and $b_L$ non-zero would just result in a total velocity

$$\mathbf{v}_{\text{trans}}(b_L, b_R) = \mathbf{v}_{\text{trans}}(b_R, b_L = 0) + \mathbf{v}_{\text{trans}}(b_L, b_R = 0),$$

i.e., the vector sum of the two velocities generated using either $b_R = 0$ or $b_L = 0$. In Fig. 5(a) and (b), we show

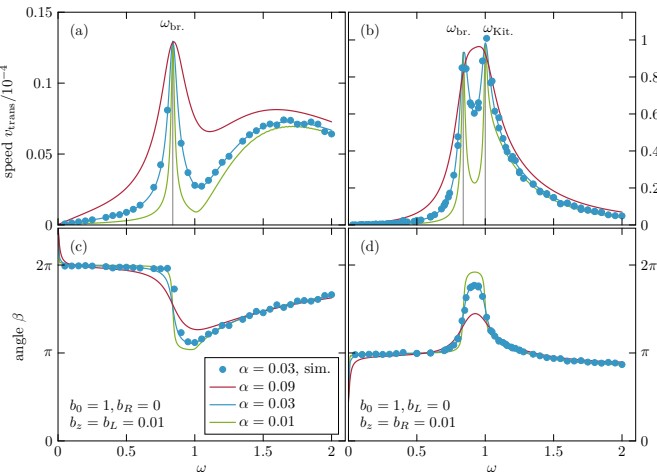

FIG. 5. Second order skyrmion translational velocity $\mathbf{v}_{\text{trans}}$ as a function of $\omega$ for $\delta, h = 0$, $\gamma < 0$, a range of $\alpha$ and a tilted driving field with $b_z = 0.01$ and either $b_L = 0.01, b_R = 0$ (left column) or $b_R = 0.01, b_L = 0$ (right column). The top row (panels (a) and (b)) shows the speed $v_{\text{trans}}$ while the bottom row (panels (c) and (d)) shows the angle $\beta$ between $\mathbf{v}_{\text{trans}}$ and $\mathbf{e}_x$. In contrast to left-polarised driving, which only resonantly excites the breathing mode, right-polarised driving resonantly excites both the breathing and Kittel modes. Varying the phase $\delta$ shifts the angle $\beta$ by $+\delta$ in panel (c) and $-\delta$ in panel (d). Similarly, setting $h = \pi/2$ (Bloch skyrmion) shifts $\beta$ down by $\pi/2$ for both panels (c) and (d). The dotted data points were obtained from numerical simulations with $\alpha = 0.03$ on mumax3.

the speed $v_{\text{trans}} = |\mathbf{v}_{\text{trans}}|$ as function of $\omega$ for the cases

$b_z = b_L = 0.01, b_R = 0$ and $b_z = b_R = 0.01, b_L = 0$, respectively. For $\gamma < 0$, the Kittel mode is only resonantly excited in the linear response when $b_R > 0$. This explains why there is only one resonance due to the breathing mode in panel (a), versus two due to the breathing and Kittel modes in panel (b). Another consequence of this effect is that $v_{\text{trans}}$ is an order of magnitude larger in panel (b) compared to panel (a). This happens because the skyrmion jellyfish does not get any help from the ferromagnetic "ocean" in the case of a left-polarised in-plane driving, so it moves more slowly. As $\alpha$ is reduced, the resonant peaks in $v_{\text{trans}}$ become better defined, while for $\omega > \omega_{\text{Kit.}}$ $v_{\text{trans}}(\omega)$ tends to a constant value. Simultaneously the transitions at $\omega \sim \omega_{\text{br.}}$ between $\beta = 2\pi$ and $\beta = \pi$ in panel (c), and similarly between $\beta = \pi$ and $\beta = 2\pi$ in panel (d), happen over a shorter range of $\omega$. Thus, following the predictions of Eq. (50) for $\omega < \omega_{\text{Kit.}}$ and in the limit of small $\alpha$, the skyrmion always travels in the $\pm \mathbf{e}_x$-directions, *except* if $\omega = \omega_{\text{br.}}$ or $\omega = \omega_{\text{Kit.}}$, in which case $\beta = 3\pi/2$ and it travels in the $-\mathbf{e}_y$-direction. We can change the angle $\beta$ by changing $\omega$, or alternatively by tuning the phase shift $\delta$ between the $b_z$ and $b_{R/L}$ components in $\mathbf{b}_1(t)$.

To check our analytical calculations we have also performed numerical simulations of the driven skyrmion using mumax3 [28, 29]. We defined the skyrmion coordinate $\mathbf{R}(t)$ to be the position where the out-of-plane magnetisation component $\hat{M}_z(\mathbf{r}, t)$ is most negative. Given that we simulate the system on a discrete lattice, a rough estimate of the skyrmion coordinate, accurate to half the lattice spacing, is provided by the lattice coordinates of the spin with the most negative $\hat{M}_z$. To improve on this, we fit the $\hat{M}_z$ components of the five spins neighbouring this spin to the right and left ($\pm \mathbf{e}_x$-direction), as well as up and down ($\pm \mathbf{e}_y$-direction), to two parabolas. The minima of these parabolas then give us much more accurate values for $\mathbf{R}(t) = (R^x(t), R^y(t))^T$. Finally, to calculate the velocity $\mathbf{v}_{\text{trans}} = \dot{\mathbf{R}}$, we measure the slope of $\mathbf{R}(t)$ at stroboscopic time intervals $\Delta t = 2\pi/\omega$ — this enables us to get rid of any periodic oscillations in $\mathbf{R}(t)$ and consequently also $\mathbf{v}_{\text{trans}}$. The resulting data, plotted as blue dots in Fig. 5, agree very nicely with our analytical calculation for the same damping parameter $\alpha = 0.03$.

## VI. CONCLUSION AND OUTLOOK

Driving a chiral magnet with a weak spatially homogeneous, time-oscillating magnetic field $\mathbf{B}_1(t)$ universally activates the translational mode(s) of the magnet. We have developed an analytical theory inspired by the effective Thiele equation which is able to precisely predict the resulting translational velocity for any magnetic texture. Within this approach, the force causing the motion is quadratic in the amplitude of the driving field, $F_{\text{trans}} \propto B_1^2$, and consists of two components, $\hat{\mathbf{M}}^{(0)} \cdot (\dot{\mathbf{M}}^{(1)} \times \nabla_i \mathbf{M}^{(1)})$ and $\alpha \dot{\mathbf{M}}^{(1)} \cdot \nabla_i \mathbf{M}^{(1)}$. Physically, the first term represents the rate of change of the local

momentum density of the magnons while the second term is a friction term, explicitly proportional to the Gilbert damping $\alpha$. We showed that the $\hat{\mathbf{M}}^{(0)} \cdot (\dot{\mathbf{M}}^{(1)} \times \nabla_i \mathbf{M}^{(1)})$ contribution vanishes from $\mathbf{F}_{\text{trans}}$ for a driven skyrmion, and in fact generally for any driven localised topological magnetic texture embedded in a ferromagnet. For driven bulk systems, $\hat{\mathbf{M}}^{(0)} \cdot (\dot{\mathbf{M}}^{(1)} \times \nabla_i \mathbf{M}^{(1)})$ is generally expected to contribute, but might also vanish for other reasons. For example, it does vanish in the driven helical and conical magnets studied in [20], but only because of the preservation of translational symmetry in the directions perpendicular to the helical pitch $\mathbf{q}$. In a bulk system with less symmetry, such as a skyrmion lattice, we would generally expect the $\hat{\mathbf{M}}^{(0)} \cdot (\dot{\mathbf{M}}^{(1)} \times \nabla_i \mathbf{M}^{(1)})$ to survive. In this case, we would expect $F_{\text{trans}}$ to be independent of $\alpha$ in the limit of low damping at all driving frequencies.

The $\alpha$-dependence of the translational velocity $\mathbf{v}_{\text{trans}}$ depends not only on $\mathbf{F}_{\text{trans}}$, but also on what happens on the left side of the Thiele equation. If the static texture is topologically trivial, $\mathbf{G} = 0$, as is the case for example in helical and conical phases, only the dissipation term $\mathcal{D}\mathbf{v}_{\text{trans}}$ survives. In this case, both sides of the Thiele equation are proportional to $\alpha$, so overall $\mathbf{v}_{\text{trans}}$ is *independent* of $\alpha$, at least when we drive non-resonantly. In the case of the skyrmion, $\mathbf{G} \neq 0$, and in the limit of low damping it will dominate the dissipation term. However, the naïve conclusion from this that $\mathbf{v}_{\text{trans}} \sim \alpha$ is wrong. The reason for this is that despite only the $\alpha\dot{\mathbf{M}}^{(1)} \cdot \nabla_i \mathbf{M}^{(1)}$ contributing, $\mathbf{F}_{\text{trans}} \sim \alpha$ only when we drive non-resonantly *below* the ferromagnetic gap, $\omega < \omega_{\text{Kit.}}$. At the breathing and Kittel resonances, or *above* the ferromagnetic gap $\omega > \omega_{\text{Kit.}}$, $\mathbf{F}_{\text{trans}}$ is in fact *independent* of $\alpha$ in the low damping limit. Consequently, $\mathbf{v}_{\text{trans}}$ is also independent of $\alpha$ at these driving frequencies. In the limit of low damping, we could thus identify two different regimes for the (non-resonantly) driven skyrmion: friction-driven if $\omega < \omega_{\text{Kit.}}$ and magnon-emission-driven if $\omega > \omega_{\text{Kit.}}$, with $v_{\text{trans}} \sim \alpha$ in the first case but independent of $\alpha$ in the second. We would expect to see similar behaviour in other driven localised magnetic defects such as skyrmion bubbles (containing no DMI), as long as these defects are topologically non-trivial. On the other hand, for defects with $Q = 0$, as is the case for example in skyrmionium [30], the situation changes dramatically. This is because the absence of $\mathbf{G} = 4\pi Q \mathbf{e}_z$ on the left side of the Thiele equation means that only the friction terms $\alpha\mathcal{D}\mathbf{v}_{\text{trans}}$ survives, so that the velocity of skyrmionium would be enhanced by a factor $1/\alpha$, making it a much better candidate than a skyrmion if the goal is to maximise $v_{\text{trans}}$. One way to enhance the skyrmion's $v_{\text{trans}}$ is to drive it near a wall. In this setting, as the skyrmion moves parallel to the wall, the gyrocoupling force $\mathbf{G} \times \mathbf{v}_{\text{trans}}$ perpendicular to the wall gets compensated by a push-back force coming from the wall. Thus the skyrmion's velocity $v_{\text{trans}}$ obtains the desired $1/\alpha$ enhancement from the surviving dissipation term.

It is also interesting to ask what happens to the system if we shake it harder, i.e., we increase the strength of the driving field $\mathbf{B}_1(t)$. Generally, we would expect our weak driving assumption $B_1/B_0 \ll 1$ to break down at some critical value of $B_1^{\text{crit}}$, after which the driven system develops dynamical instabilities. In the driven helical phases studied in [20], we saw that the leading order instabilities are Floquet magnon "laser" instabilities, and $B_1^{\text{crit.}} \sim \alpha$ directly depends on the amount of damping in the system. In the presence of these instabilities, the driven system becomes a kind of time quasicrystal, with macroscopic occupation of a magnon state whose frequency and momentum are incommensurate with the driving frequency $\omega$ and lattice momentum $\mathbf{q}$, respectively. This effect is in fact universal to any periodically driven bulk system with lattice symmetry, and we would also expect to see it for example in a driven skyrmion lattice. It is less easy to predict what the leading order instability would be in a localised system such as the driven single skyrmion. If it were still the Floquet magnon mechanism, we would expect a range of scattering $k$ magnons to become unstable, rather than just a single one, due to the lack of discrete lattice symmetry of the single skyrmion. Another option is that the ferromagnetic bulk breaks down, and the skyrmion serves as a kind of "seed" for the formation of a skyrmion lattice — see also [31], where the "seeds" instead took the form of artificial rectangular holes in the sample. To test what happens, we have performed some preliminary numerical simulations, keeping the driving frequency near the breathing resonance, $\omega = 0.83$, with $b_0 = 1, \alpha = 0.01, b_L = 0$ also fixed, and increasing $b_z = b_R$ between 0.001 and 0.1. The results of these simulations show that there is first a transition from the expected $v_{\text{trans}} \sim b_z b_R$ predicted by our weak driving field theory to $v_{\text{trans}} \sim \sqrt{b_z b_R}$ at $b_z^{\text{crit}}, b_R^{\text{crit}} \sim 0.01$. Then, at $b_z = b_R \sim 0.05$, the skyrmion disappears, with the system's topological charge $Q$ going from $-1$ to 0. However, before making any hasty conclusions about the fate of the driven skyrmion as we increase $B_1$, further numerical experimentation at different driving frequencies and lattice discretisations is required, as these factors can greatly influence what happens to the driven system.

We now wish to make an experimental prediction for the speed $V_{\text{trans}}$ of the driven skyrmion. We do this for the metallic chiral magnet MnSi, whose micromagnetic parameters are $\gamma = -1.76 \times 10^{-11}$ $\text{T}^{-1}\text{s}^{-1}$, $J = 7.05 \times 10^{-13}$ $\text{Jm}^{-1}$, $D = 2.46 \times 10^{-4}$ $\text{Jm}^{-2}$, $M_0 = 1.52 \times 10^5$ $\text{Am}^{-1}$ [32–34] and $\alpha \sim 0.01$. Using the results of our amplitude run, we assume that $b_1^{\text{crit}} \sim b_z^{\text{crit}} \sim b_R^{\text{crit}} \sim 0.01$, which translates into $B_1^{\text{crit}} \sim 5\text{mT}$ in physical units. This is within an order of magnitude of $B_1^{\text{crit}}$ for driven helical magnets, which we calculated to be $0.5\text{mT}$ on resonance [20]. Using the parameters of a tilted driving field with $B_z \sim B_x \sim 5$ mT and $\Omega \sim 100$ GHz, we predict the skyrmion to reach speeds of $V_{\text{trans}} \sim 30$ mms$^{-1}$. This is over two orders of magnitude *larger* than the minimum depinning velocity for skyrmions, estimated from

measurements of the Hall effect to be 0.2 mms$^{-1}$ for a critical current density $2j_c$ [12]. Thus, it should be easy to observe the skyrmion jellyfish experimentally, at least in materials with low pinning such as MnSi. In the real world all systems will also carry a degree of disorder. Far from hindering the skyrmion motion, this might actually have an enhancing effect on its speed $V_{\text{trans}}$. A driven skyrmion lattice will start to rotate via a mechanism similar to the one we presented here for the translation motion. Recent experiments where skyrmion lattices in the insulator $Cu_2OSeO_3$ were driven by femtosecond lasers pulses resulted in rotational speeds up to $2 \times 10^7$–$10^8$ deg s$^{-1}$, which is over six order of magnitude *faster* than predicted theoretically using our theory for a clean system. Finding a way to incorporate disorder into our theoretical model is therefore crucial to gaining better predictive power for future experiments.

To conclude, we have presented a fully analytical theory for realising a skyrmion jellyfish by driving a chiral magnet with oscillating magnetic fields. We hope these insights will be of interest to other fans of nano-scale man-made marine life.

## ACKNOWLEDGMENTS

We thank Joachim Hemberger, Christian Pfleiderer, Volodymyr Kravchuk, Markus Garst and especially Achim Rosch for useful discussions and guidance. We acknowledge the financial support of the DFG via SPP 2137 (project number 403505545) and CRC 1238 (project number 277146847, subproject C04). We also thank the Regional Computing Center of the University of Cologne (RRZK) for providing computing time on the DFG-funded (Funding number: INST 216/512/1FUGG) High Performance Computing (HPC) system CHEOPS as well as technical support. The highly accurate shooting numerics for the finite-$k$ scatttering modes was implemented using the `BigFloat` type and the excellent `DifferentialEquations.jl` library [22], which has the `RK065` algorithm readily available to deal with the singularity at $r = 0$.

## AUTHOR CONTRIBUTIONS

NdS designed the study, performed the analytical and numerical calculations and wrote the paper. VL provided technical support with implementing the eigenbasis.

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

## Appendix A: Supplementary videos

The supplementary videos `first_order_bZ.mp4` and `first_order_bR.mp4` show the response of the magnetisation up to $\mathcal{O}(\epsilon^1)$, $\hat{\mathbf{M}}^{(0)}(\mathbf{r}) + \mathbf{M}^{(1)}(\mathbf{r}, t)$, to driving fields $b_z \cos(\omega t)\mathbf{e}_z$ and $b_R (\cos(\omega t)\mathbf{e}_x + \sin(\omega t)\mathbf{e}_y)$, respectively. In `first_order_bZ.mp4`, the radial symmetry of the skyrmion is preserved at all times, as only the $m = 0$ angular momentum sector is excited. This is no longer the case in `first_order_bR.mp4`, where the $m = \pm 1$ angular momentum sectors are excited. In `second_order_tilted_drive.mp4`, we show the response including the second order translational motion, $\hat{\mathbf{M}}^{(0)}(\mathbf{r} - \mathbf{v}_{\text{trans}}t) + \mathbf{M}^{(1)}(\mathbf{r}, t)$ to a tilted driving field $\mathbf{b}_1(t) = b_R (\cos(\omega t)\mathbf{e}_x + \sin(\omega t)\mathbf{e}_y) + b_z \cos(\omega t)\mathbf{e}_z$. The skyrmion starts to "swim" as a result of its periodic first order asymmetric contractions and relaxations — just like a jellyfish.

## Appendix B: Activation of Translational Modes – auxiliary calculations

### 1. $\nabla_i \hat{\mathbf{M}} \cdot (\hat{\mathbf{M}} \times \mathbf{LLG})$ projection

Taking $\hat{\mathbf{M}} \times$ Eq. (7) and multiplying everything by $\text{sgn}(\gamma)/M_0$, we obtain

$$\text{sgn}(\gamma)\hat{\mathbf{M}} \times \dot{\hat{\mathbf{M}}} = |\gamma| \left( (\hat{\mathbf{M}} \cdot \mathbf{B}_{\text{eff}})\hat{\mathbf{M}} - \mathbf{B}_{\text{eff}} \right) + \alpha\dot{\hat{\mathbf{M}}}, \quad \text{(B1)}$$

where we used $\hat{\mathbf{M}} \cdot \dot{\hat{\mathbf{M}}} = \frac{1}{2}\frac{d}{dt}\left(\hat{\mathbf{M}} \cdot \hat{\mathbf{M}}\right) = 0$. Next, we project Eq. (B1) onto $\nabla_i\hat{\mathbf{M}}$, $i = \{x, y, z\}$, and then integrate over 3D space,

$$\text{sgn}(\gamma) \int \mathrm{d}^3r \, \nabla_i\hat{\mathbf{M}} \cdot \left( \hat{\mathbf{M}} \times \dot{\hat{\mathbf{M}}} \right) = \alpha \int \mathrm{d}^3r \, \nabla_i\hat{\mathbf{M}} \cdot \dot{\hat{\mathbf{M}}}, \quad \text{(B2)}$$

where we got rid of the two first terms on the right of Eq. (B1) using $\hat{\mathbf{M}} \cdot \nabla_i\hat{\mathbf{M}} = \frac{1}{2}\nabla_i\left(\hat{\mathbf{M}} \cdot \hat{\mathbf{M}}\right) = 0$ and

$$\int \mathrm{d}^3r \, \mathbf{B}_{\text{eff}} \cdot \nabla_i\hat{\mathbf{M}} = -\int \mathrm{d}^3r \, \frac{\delta F}{\delta\mathbf{M}} \cdot \frac{d\hat{\mathbf{M}}}{dr_i} = -\frac{1}{M_0}\nabla_i F = 0, \quad \text{(B3)}$$

as the free energy is translationally invariant. Finally, substituting Eq. (11) into Eq. (B2), collecting all terms proportional to $\epsilon^2$ and keeping only the DC component of the resulting equation, we obtain Eq. (12).

### 2. $\nabla_i\hat{\mathbf{M}}^{(0)} \cdot (\hat{\mathbf{M}}^{(0)} \times \mathbf{LLG})$ projection

This time we act with $\nabla_i\hat{\mathbf{M}}^{(0)} \cdot (\hat{\mathbf{M}}^{(0)} \times$, i.e. *only* the $\mathcal{O}(\epsilon^0)$ components of $\hat{\mathbf{M}} \cdot (\hat{\mathbf{M}} \times$, on Eq. (7). We are still interested in the $\mathcal{O}(\epsilon^2)$ contribution at the end, which can now only come from the LLG terms. For the $\dot{\hat{\mathbf{M}}}$ and

$\alpha\hat{\mathbf{M}} \times \dot{\hat{\mathbf{M}}}$ terms, these contributions are

$$\int \mathrm{d}^3r\nabla_i\hat{\mathbf{M}}^{(0)} \cdot (\hat{\mathbf{M}}^{(0)} \times \dot{\hat{\mathbf{M}}}^{(2)}) = -(\mathbf{G} \times \mathbf{v}_{\text{trans}})_i,$$

$$\alpha \int \mathrm{d}^3r\nabla_i\hat{\mathbf{M}}^{(0)} \cdot (\hat{\mathbf{M}}^{(0)} \times (\hat{\mathbf{M}}^{(0)} \times \dot{\hat{\mathbf{M}}}^{(2)})) = \alpha(\mathcal{D}\mathbf{v}_{\text{trans}})_i,$$

where we used $\hat{\mathbf{M}}^{(0)} \perp \mathbf{M}^{(1)}, \dot{\mathbf{M}}^{(1)}$ to get rid of the $\hat{\mathbf{M}}^{(0)} \times (\mathbf{M}^{(1)} \times \dot{\mathbf{M}}^{(1)})$ term in the dissipation term. Thus, the $\mathcal{O}(\epsilon^2)$ contributions coming from the gyrocoupling and dissipation term of the LLG are actually *simpler* using this method of projection, compared to App. B.1.

As always though, nothing is for free. In return for simpler gyrocoupling and dissipation contributions, we get extra terms coming from $\mathbf{B}_{\text{eff}}$. While previously this term integrated out to zero (see Eq. (B3)), it now produces finite $\mathcal{O}(\epsilon^2)$ contributions,

$$\int \mathrm{d}^3r\nabla_i\hat{\mathbf{M}}^{(0)} \cdot \left( \hat{\mathbf{M}}^{(0)} \times (\hat{\mathbf{M}} \times \mathbf{B}_{\text{eff}}) \right) =$$

$$\int \mathrm{d}^3r \left[ (\nabla_i\hat{\mathbf{M}}^{(0)} \cdot \mathbf{M}^{(1)})(\hat{\mathbf{M}}^{(0)} \cdot \mathbf{B}_{\text{eff}}^{(1)}) \right.$$

$$\left. + (\hat{\mathbf{M}}^{(0)} \cdot \mathbf{B}_{\text{eff}}^{(0)})(\nabla_i\hat{\mathbf{M}}^{(0)} \cdot \mathbf{M}^{(2)}) - \nabla_i\hat{\mathbf{M}}^{(0)} \cdot \mathbf{B}_{\text{eff}}^{(2)} \right], \quad \text{(B4)}$$

where we used $\hat{\mathbf{M}}^{(0)} \perp \mathbf{M}^{(1)}$ and $\hat{\mathbf{M}}^{(0)} \parallel \mathbf{B}_{\text{eff}}^{(0)} \perp \nabla_i\hat{\mathbf{M}}^{(0)}$. We need to check what happens to the $\mathbf{M}_{\text{stat.}}^{(2)}$ in Eq. (B4), as its presence in the integral would make our new approach unusable for evaluating $\mathbf{v}_{\text{trans}}$. For this, it is helpful to split $\mathbf{M}_{\text{stat.}}^{(2)}$ into components parallel and perpendicular to $\hat{\mathbf{M}}^{(0)}$,

$$\mathbf{M}_{\text{stat.}}^{(2)} = w_\parallel\hat{\mathbf{M}}^{(0)} + w_\perp\mathbf{e}_- + w_\perp^*\mathbf{e}_+,$$

where the $w_\parallel$, $w_\perp$ are complex coefficients to be determined. Using the normalisation condition $\langle\hat{\mathbf{M}} \cdot \hat{\mathbf{M}}\rangle_t = 1$ to all orders of $\epsilon$, we see that the $w_\parallel$ component is already defined as a function of the $\mathcal{O}(\epsilon^1)$ fields,

$$w_\parallel = -\frac{1}{2}\langle\mathbf{M}^{(1)} \cdot \mathbf{M}^{(1)}\rangle_t,$$

where we used the expansion of $\hat{\mathbf{M}}$ given in Eq. (9). Next, we will show that the $w_{\perp,\pm}$ components, which we have not calculated, drop out of Eq. (B4), and only $w_\parallel$ remains. Integrating the last term on the RHS of Eq. (B4) by parts, we have

$$\int \mathrm{d}^3r\nabla_i\hat{\mathbf{M}}^{(0)} \cdot \mathbf{B}_{\text{eff}}^{(2)}$$

$$= \int \mathrm{d}^3r\nabla_i\hat{\mathbf{M}}^{(0)} \cdot \left( \tilde{J}\nabla^2 - 2\tilde{D}\boldsymbol{\nabla}\times \right) \mathbf{M}^{(2)}$$

$$= \int \mathrm{d}^3r\mathbf{M}^{(2)} \cdot \left( \tilde{J}\nabla^2 - 2\tilde{D}\boldsymbol{\nabla}\times \right) \nabla_i\hat{\mathbf{M}}^{(0)}$$

$$= \int \mathrm{d}^3r\mathbf{M}^{(2)} \cdot \nabla_i\mathbf{B}_{\text{eff}}^{(0)}$$

$$= \int \mathrm{d}^3r\mathbf{M}^{(2)} \cdot \nabla_i \left( (\hat{\mathbf{M}}^{(0)} \cdot \mathbf{B}_{\text{eff}}^{(0)})\hat{\mathbf{M}}^{(0)} \right),$$

where $\tilde{J} = J/M_0$, $\tilde{D} = D/M_0$. We assumed that the surface terms in the integration by parts step vanish. This assumption is valid when $\nabla_i \mathbf{M}^{(0)}$ is bounded, as for example it is in the case of a single skyrmion, but not in general. In the penultimate step we switched the orders of the $\nabla_i$ and $\tilde{J}\nabla^2 - 2\tilde{D}\boldsymbol{\nabla}\times$ operators, and in the last step we used $\hat{\mathbf{M}}^{(0)} \parallel \mathbf{B}_{\text{eff}}^{(0)}$. Inserting this into Eq. (B4) and simplifying, the only remaining term carrying $\mathbf{M}^{(2)}$ reads

$$-\int \mathrm{d}^3 r \hat{\mathbf{M}}^{(0)} \cdot \mathbf{M}^{(2)} \nabla_i \left( \hat{\mathbf{M}}^{(0)} \cdot \mathbf{B}_{\text{eff}}^{(0)} \right). \qquad \text{(B5)}$$

Substituting

$$\mathbf{M}^{(2)} = -t\mathbf{v}_{\text{trans}} \cdot \nabla \hat{\mathbf{M}}^{(0)} + \mathbf{M}_{\text{stat.}}^{(2)},$$

valid for short $t$, into Eq. (B5), we see that only the $w_\parallel$ term survives, giving

$$\frac{1}{2} \int \mathrm{d}^3 r \langle \mathbf{M}^{(1)} \cdot \mathbf{M}^{(1)} \rangle_t \nabla_i \left( \hat{\mathbf{M}}^{(0)} \cdot \mathbf{B}_{\text{eff}}^{(0)} \right).$$

Putting this all together we obtain Eq. (16).

### Appendix C: LLG in the language of $T_{\mu\nu}$

#### 1. Lagrangian density $\mathcal{L}$, $T_{\mu\nu}$ and divergence of $T_{\mu\nu}$

The total Lagrangian density of the chiral magnet modelled by Eq. (1) is

$$\mathcal{L} = \mathcal{L}^{\text{dyn}} + \mathcal{L}^{\text{stat}}, \qquad \text{(C1)}$$

$$\mathcal{L}^{\text{dyn}} = \text{sgn}(\gamma)\mathbf{A} \cdot \dot{\hat{\mathbf{M}}},$$

$$\mathcal{L}^{\text{stat}} = \frac{1}{2}\nabla_\mu \hat{\mathbf{M}} \cdot \nabla_\mu \hat{\mathbf{M}} + \hat{\mathbf{M}} \cdot (\boldsymbol{\nabla} \times \hat{\mathbf{M}}) - \mathbf{b}_{\text{ext}} \cdot \hat{\mathbf{M}},$$

where the gauge field $\mathbf{A}(\hat{\mathbf{M}})$ obeys $\frac{\partial A_k}{\partial \hat{\mathbf{M}}_j} - \frac{\partial A_j}{\partial \hat{\mathbf{M}}_k} = \epsilon_{ijk}\hat{\mathbf{M}}_i$. Note that in Eq. (C1), we are using dimensionless space, time and magnetic field units, defined as $\tilde{r}_i = (D/J)r_i$, $\tilde{t} = D^2|\gamma|/(JM_0)t$ and $b_i = M_0 J/D^2 B_i$, respectively, and immediately dropping the tildes on $\tilde{r}_i$, $\tilde{t}$ for a cleaner notation. The stress energy tensor resulting from translational symmetry is

$$T_{\mu\nu} = \frac{\partial \mathcal{L}}{\partial(\partial_\mu \hat{\mathbf{M}})} \cdot \partial_\nu \hat{\mathbf{M}} - \delta_{\mu\nu}\mathcal{L}, \qquad \text{(C2)}$$

where $\mu, \nu = \{t, x, y, z\}$. To take into account the phenomenological damping, the divergence of $T_{\mu\nu}$ needs to be updated to include a term proportional to $\alpha$ on the RHS,

$$\partial_\mu T_{\mu\nu} = \alpha\dot{\hat{\mathbf{M}}} \cdot \partial_\nu \hat{\mathbf{M}}. \qquad \text{(C3)}$$

The LLG is then obtained from the spatial components $\nu = \{x, y, z\}$ of Eq. (C3). We can use Eq. (C3) to define some momentum and current densities. We define

the rate of change of the momentum density purely as a function of $\mathcal{L}^{\text{dyn}}$,

$$\dot{P}_\nu = \frac{\partial}{\partial t}\left( \frac{\partial \mathcal{L}^{\text{dyn}}}{\partial \dot{\hat{\mathbf{M}}}} \cdot \nabla_\nu \hat{\mathbf{M}} \right) - \nabla_\nu \left( \mathcal{L}^{\text{dyn}} \right)$$

$$= \text{sgn}(\gamma)\hat{\mathbf{M}} \cdot \left( \dot{\hat{\mathbf{M}}} \times \nabla_\nu \hat{\mathbf{M}} \right). \qquad \text{(C4)}$$

This allows us to define the current densities purely in terms of $\mathcal{L}^{\text{stat}}$ as

$$(\mathbf{J}_\nu)_i = \frac{\partial \mathcal{L}^{\text{stat}}}{\partial \nabla_i \hat{\mathbf{M}}} \cdot \nabla_\nu \hat{\mathbf{M}} - \delta_{i\nu}\mathcal{L}^{\text{stat}} \qquad \text{(C5)}$$

It is not too difficult to show that the divergence of these current densities is given by

$$\nabla \cdot \mathbf{J}_\nu = \mathbf{b}_{\text{eff}} \cdot \nabla_\nu \hat{\mathbf{M}}, \qquad \text{(C6)}$$

where $\mathbf{b}_{\text{eff}}$ was defined in Eq. (7). Also, when $r \gtrsim l$ so that all finite $k$ magnons have decayed, $\hat{\mathbf{M}}$ becomes constant in space. The only surviving term in $\mathbf{J}_\nu$ in this case comes from the $-\mathbf{b}_{\text{ext}} \cdot \hat{\mathbf{M}}$ in $\mathcal{L}$, so that

$$\mathbf{J}_\nu = (\mathbf{b}_{\text{ext}} \cdot \hat{\mathbf{M}})\mathbf{e}_\nu.$$

The continuity equation updated to include Gilbert damping reads

$$\dot{P}_\nu + \nabla \cdot \mathbf{J}_\nu = \alpha\dot{\hat{\mathbf{M}}} \cdot \nabla_\nu \hat{\mathbf{M}}. \qquad \text{(C7)}$$

#### 2. Eq. (C7) to order $\mathcal{O}(\epsilon^2)$

At second order in $\epsilon$, $\dot{P}_\nu^{(2)}$ and $(\dot{\hat{\mathbf{M}}} \cdot \nabla_\nu \hat{\mathbf{M}})^{(2)}$ will have two kinds of terms which either include or don't include $\mathbf{v}_{\text{trans}}$. It is straightforward to show that the terms which survive after time averaging over $T$ are

$$\langle \dot{P}_\nu^{(2)} \rangle_t = -\text{sgn}(\gamma)(\mathbf{q}^{\text{top}} \times \mathbf{v}_{\text{trans}})_\nu + \langle \dot{P}_\nu^{\text{m}} \rangle_t$$

$$\langle (\dot{\hat{\mathbf{M}}} \cdot \nabla_\nu \hat{\mathbf{M}})^{(2)} \rangle_t = -(\mathfrak{D}\mathbf{v}_{\text{trans}})_\nu + \langle \dot{\mathbf{M}}^{(1)} \cdot \nabla_\nu \mathbf{M}^{(1)} \rangle_t.$$

where $\mathbf{q}^{\text{top}}$ and $\mathfrak{D}$ are just the local topological charge and dissipation matrix densities of the static texture,

$$q_\alpha^{\text{top}} = \frac{1}{2}\epsilon_{\alpha\beta\gamma}\hat{\mathbf{M}}^{(0)} \cdot (\nabla_\beta \hat{\mathbf{M}}^{(0)} \times \nabla_\gamma \hat{\mathbf{M}}^{(0)})$$

$$\mathfrak{D}_{\alpha\beta} = \nabla_\alpha \hat{\mathbf{M}}^{(0)} \cdot \nabla_\beta \hat{\mathbf{M}}^{(0)} \qquad \text{(C8)}$$

and $\langle \dot{P}_\nu^{\text{m}} \rangle$ is the time-averaged rate of change of the magnon momentum density,

$$\dot{P}_\nu^{\text{m}} = \text{sgn}(\gamma)\hat{\mathbf{M}}^{(0)} \cdot \left( \dot{\mathbf{M}}^{(1)} \times \nabla_\nu \mathbf{M}^{(1)} \right). \qquad \text{(C9)}$$

We also need to calculate the $\nabla \cdot \mathbf{J}_\nu$ to order $\mathcal{O}(\epsilon^2)$. This should not carry any contribution from $\mathbf{v}_{\text{trans}}t$, as the lack of a time derivative in $\mathbf{J}_\nu$ would result in such a term exploding for large $t$. For short times, the contribution

to $\hat{\mathbf{M}}^{(2)}$ coming from $\mathbf{v}_{\text{trans}}$ is $-t(\mathbf{v}_{\text{trans}} \cdot \nabla)\hat{\mathbf{M}}^{(0)}$. Using this and Eq. (C6), we can write down the contributions to $\nabla \cdot \mathbf{J}_\nu$ coming form $\mathbf{v}_{\text{trans}}$ as

$$
-t(\mathbf{v}_{\text{trans}})_\alpha \left( \mathbf{b}_{\text{eff}}^{(0)} \cdot \nabla_\nu \nabla_\alpha \hat{\mathbf{M}}^{(0)} + \nabla_\nu \hat{\mathbf{M}}^{(0)} \cdot \nabla_\alpha \mathbf{b}_{\text{eff}}^{(0)} \right)
$$
$$
= -t\mathbf{v}_{\text{trans}} \cdot \nabla \left( \mathbf{b}_{\text{eff}}^{(0)} \cdot \nabla_\nu \hat{\mathbf{M}}^{(0)} \right) = 0.
$$

where we used $\mathbf{b}_{\text{eff}}^{(0)} \parallel \hat{\mathbf{M}}^{(0)}$ and $\hat{\mathbf{M}}^{(0)} \cdot \nabla_\nu \hat{\mathbf{M}}^{(0)} = 0$ in the last line. While $\nabla \cdot \mathbf{J}_\nu$ does not carry any terms proportional to $\mathbf{v}_{\text{trans}}$, it will generally have contributions from the other non temporally oscillating static second order term, $\mathbf{M}_{\text{stat.}}^{(2)}$.

The analysis we have performed enables us to separate the $\mathbf{v}_{\text{trans}}$ terms from the other terms in Eq. (C7), giving

$$
-\operatorname{sgn}(\gamma) \left( \mathbf{q}^{\text{top}} \times \mathbf{v}_{\text{trans}} \right)_\nu + \alpha \left( \mathfrak{D}\mathbf{v}_{\text{trans}} \right)_\nu = \mathfrak{F}_\nu^{\text{tot}}, \quad \text{(C10)}
$$
$$
\mathfrak{F}_\nu^{\text{tot}} = -\langle \dot{P}_\nu^{\text{m}} \rangle_t - \langle \nabla \cdot \mathbf{J}_\nu^{(2)} \rangle_t + \alpha \langle \dot{\mathbf{M}}^{(1)} \cdot \nabla_\nu \mathbf{M}^{(1)} \rangle_t.
$$

In its given form, Eq. (C10) is actually useless for calculating $\mathbf{v}_{\text{trans}}$ because of the presence of the unknown component $\mathbf{M}_{\text{stat.}}^{(2)}$ in $\langle \nabla \cdot \mathbf{J}_\nu \rangle_t$. To obtain $\mathbf{v}_{\text{trans}}$, we would have to integrate Eq. (C10) over all space to make $\nabla \cdot \mathbf{J}_\nu$ vanish, which just returns Eq. (12). Nevertheless, Eq. (C10) is useful for investigating which of $\langle \dot{P}_\nu^{\text{m}} \rangle_t$, $\langle \nabla \cdot \mathbf{J}_\nu \rangle_t$ or $\alpha \langle \dot{\mathbf{M}}^{(1)} \cdot \nabla_\nu \mathbf{M}^{(1)} \rangle_t$ plays a dominant role in the local force density.

### Appendix D: Equation of motion for $a, a^*$

Only two kinds of terms survive the projection of Eq. (20) onto $\mathbf{e}_\pm$: $\mathbf{e}_\pm \cdot \mathbf{e}_\pm = 1$ and $\mathbf{e}_\pm \cdot (\mathbf{e}_3 \times \mathbf{e}_\mp) = \pm i$. This results in the following equations of motion

$$
\operatorname{sgn}(\gamma) \frac{d}{dt} (as) = i \{F, as\}
$$
$$
- i\alpha \left( (1 - a^*a) \frac{d}{dt} (as) - \frac{d}{dt}(a^*a)as \right),
$$
$$
\operatorname{sgn}(\gamma) \frac{d}{dt} (a^*s) = i \{F, a^*s\} \qquad \text{(D1)}
$$
$$
+ i\alpha \left( (1 - a^*a) \frac{d}{dt} (a^*s) - \frac{d}{dt}(a^*a)a^*s \right),
$$

where $s = \sqrt{1 - \frac{a^*a}{2}}$, correct to all orders in $a, a^*$.

### Appendix E: Expressions for $F^{(1)}$ and $F^{(2)}$

To find $F^{(1)}$ and $F^{(2)}$, we substitute Eq. (17) into Eq. (1) and Taylor expand in $a, a^*$. To avoid cluttering the expressions with too many constants, all the free energies we list below are rescaled via $\tilde{F} = (D/J^2)F$. We also use a dimensionless length scale $\tilde{r}_i = (D/J)r_i$, implying $\tilde{\nabla}_i = (J/D)\nabla_i$ for the spatial gradients appearing in the Heisenberg and DMI energy terms. For readability we then drop the tildes on both $\tilde{F}$ and $\tilde{r}$.

In Eq. (5), we already introduced a rescaled dimensionless amplitude $b_0$ for the static component of the external magnetic field. We now do the same for the amplitudes of the oscillating components, defining $b_i = (M_0 J/D^2)B_1^i$, $i = \{x, y, z\}$. It is more natural to rewrite $b_x, b_y$, the oscillating field components in the plane of the skyrmion, in terms of the circularly polarised driving field components

$$
\begin{aligned}
b_R &= b_x + b_y, \\
b_L &= b_x - b_y.
\end{aligned} \qquad \text{(E1)}
$$

Using polar coordinates, the free energy $F$ and free energy density $\mathcal{F}$ are related via $F = \int_0^\infty r\,dr \int_0^{2\pi} d\chi\, \mathcal{F}$. With this definition, we obtain the free energy densities

$$
\begin{aligned}
\mathcal{F}_{\text{drive}}^{(1)} = \frac{\epsilon}{\sqrt{2}} \Big[ & (a + a^*)\Big( b_z \sin(\theta_0) \cos(\omega t + \delta) \\
& - \frac{1}{2} \cos(\theta_0) \left( b_R \cos(\phi - \omega t) + b_L \cos(\phi + \omega t) \right) \Big) \\
& - \frac{i}{2}(a - a^*) \left( b_R \sin(\phi - \omega t) + b_L \sin(\phi + \omega t) \right) \Big],
\end{aligned} \qquad \text{(E2)}
$$

$$
\begin{aligned}
\mathcal{F}_{\text{drive}}^{(2)} = \frac{\epsilon}{2} a^* a \Big[ & 2b_z \cos(\theta_0) \cos(\omega t + \delta) \\
& + \sin(\theta_0) \left( b_R \cos(\phi - \omega t) + b_L \cos(\phi + \omega t) \right) \Big].
\end{aligned} \qquad \text{(E3)}
$$

$\mathcal{F}_{\text{no drive}}^{(1)} = 0$, otherwise the skyrmion texture would be moving, rather than static, in the absence of a driving field $\mathbf{b}_1(t)$. The lowest non-zero contribution is therefore quadratic in $a, a^*$, and given by

$$
\begin{aligned}
\mathcal{F}_{\text{no drive}}^{(2)} = -\frac{1}{4r^2} \Big[ & \\
& + aa^*\Big( -4b_0 r^2 \cos(\theta_0) + 2r^2\theta_0'^2 + 4r^2\theta_0' \\
& + 6r \sin(2\theta_0) - 3\cos(2\theta_0) - 1 \Big) \\
& + 4i(a^* \partial_\chi a - a\partial_\chi a^*)(\cos(\theta_0) - r\sin(\theta_0)) \\
& - 4\left( \partial_\chi a \partial_\chi a^* + r^2 a' a^{*'} \right) \\
& + (a^2 + a^{*2})\Big( -r^2\theta_0'^2 - 2r^2\theta_0' \\
& + \sin^2(\theta_0) + r\sin(2\theta_0) \Big) \Big].
\end{aligned} \qquad \text{(E4)}
$$

### Appendix F: force $f$ and matrix $H_m$

The force $f(t)$ is given by $\{F_{\text{drive}}^{(1)}, a\}$,

$$
\begin{aligned}
f(t) = \frac{1}{2\sqrt{2}} \Big( & b_z \sin(\theta_0) \left( e^{i(\omega t + \delta)} + e^{-i(\omega t + \delta)} \right) \\
& - \frac{1}{2} \left( (\cos(\theta_0) - 1) \left( b_R e^{i(\chi + h - \omega t)} + b_L e^{i(\chi + h + \omega t)} \right) \right) \\
& - \frac{1}{2} \left( (\cos(\theta_0) + 1) \left( b_R e^{-i(\chi + h - \omega t)} + b_L e^{-i(\chi + h + \omega t)} \right) \right) \Big).
\end{aligned} \qquad \text{(F1)}
$$

To calculate $H_m$, we evaluate the Poisson brackets $\{F_{\text{no drive}}^{(2)}, a\}$, $\{F_{\text{no drive}}^{(2)}, a^*\}$, which include a term

$$\int r \, dr \, \partial_r a_m(r) \{\partial_r a_m^*(r), a_m(r')\} =$$

$$\int r \, dr \, \partial_r a_m(r) \partial_r \{a_m^*(r), a_m(r')\} = \frac{1}{r'} \partial_{r'}(r' a_m(r')).$$

$H_m$ can then be written in the form

$$H_m = \mathbb{1} \left( -\partial_r^2 - \frac{1}{r}\partial_r + \frac{m^2+1}{r^2} + b_0 + V_0 \right) \quad \text{(F2)}$$

$$+ \sigma^z \frac{2m}{r^2} + \sigma^z V_z^m + \sigma^x V_x,$$

$$V_0 = \frac{3\left(\cos(2\theta_0) - 1\right)}{4r^2} - \frac{3\sin(2\theta_0)}{2r}$$

$$+ b_0\left(\cos(\theta_0) - 1\right) - \theta_0' - \frac{\theta_0'^2}{2},$$

$$V_z^m = \frac{2m}{r^2}\left(\cos(\theta_0) - 1 - r\sin(\theta_0)\right),$$

$$V_x = -\frac{1}{2r^2}\left(\sin^2(\theta_0) + r\sin(2\theta_0) - r^2\theta_0'^2 - 2r^2\theta_0'\right),$$

where the potentials $V_0$, $V_x$, $V_z^m$ vanish for $r \gg r_0$.

## Appendix G: Damped eigenbasis of $\sigma^z H_m$

### 1. Particle-hole property of $\left|m, k^{(0)}\right\rangle$

Taking the complex conjugate of Eq. (29) and using the property $\sigma^x H_m \sigma^x = H_{-m}$, we have

$$E_{m,k,i}^* \left(\text{sgn}(\gamma) - i\alpha\sigma^z\right) |m, k, i\rangle^* = \sigma^z \sigma^x H_{-m} \sigma^x |m, k, i\rangle^*$$

Pre-multiplying this equation with $\sigma^x$, and using $\sigma^x \sigma^z = -\sigma^z \sigma^x$, we obtain

$$-E_{m,k,i}^* \left(\text{sgn}(\gamma) + i\alpha\sigma^z\right) \sigma^x |m, k, i\rangle^* = \sigma^z H_{-m} \sigma^x |m, k, i\rangle^*.$$

Thus, $\sigma^x |m, k, i\rangle^*$ is also an eigenvector of $\sigma^z H_{-m}$ with eigenvalue $-E_{m,k,i}^*$.

### 2. Inner products between $\left|m, k^{(0)}\right\rangle$

For each $m$-sector, the scattering states obey

$$\left\langle m, k^{(0)} \left| \sigma^z \right| m, k'^{(0)} \right\rangle = \frac{\delta(k - k')}{k},$$

$$\left\langle m, k^{(0)} \left| \sigma^z \sigma^x \right| m, k^{(0)} \right\rangle = 0.$$

In addition, we have for the $m = 0$ sector,

$$\left\langle 0, \text{br.}^{(0)} \left| \sigma^z \right| 0, \text{br.}^{(0)} \right\rangle = 1,$$

$$\left\langle 0, \text{br.}^{(0)} \left| \sigma^z \sigma^x \right| 0, \text{br.}^{(0)} \right\rangle = 0,$$

$$\left\langle 0, \text{br.}^{(0)} \left| \sigma^z \right| 0, k^{(0)} \right\rangle = 0,$$

$$\left\langle 0, \text{br.}^{(0)} \left| \sigma^z \sigma^x \right| 0, k^{(0)} \right\rangle = 0.$$

And finally for the $m = \pm 1$ sectors,

$$\left\langle \pm 1, \text{trans.}^{(0)} \left| \sigma^z \right| \pm 1, \text{trans.}^{(0)} \right\rangle = \pm 1,$$

$$\left\langle \pm 1, \text{trans.}^{(0)} \left| \sigma^z \right| \pm 1, k^{(0)} \right\rangle = 0.$$

### 3. First order perturbation theory in $\alpha$

Substituting Eq. (30) into Eq. (29) and keeping only the linear in $\alpha$ terms, we obtain the equation

$$(\epsilon_{m,k}^{(0)} \sigma^z - \epsilon_{m,k}^{(1)}) \left|m, k^{(0)}\right\rangle + \epsilon_{m,k}^{(0)} \left|m, k^{(1)}\right\rangle = \sigma^z H_m \left|m, k^{(1)}\right\rangle. \quad \text{(G1)}$$

Projecting $\left\langle m, k^{(0)} \right| \sigma^z$ onto Eq. (G1) and using $\epsilon_{m,k}^{(0)} \left\langle m, k^{(0)} \right| \sigma^z = \left\langle m, k^{(0)} \right| H_m$, we obtain the $\mathcal{O}(\alpha)$ corrections to the energies,

$$\epsilon_{\text{br.}}^{(1)} = \left\langle 0, \text{br.}^{(0)} \middle| 0, \text{br.}^{(0)} \right\rangle \epsilon_{\text{br.}}^{(0)}, \quad \epsilon_{\text{trans.}}^{(1)} = 0, \quad \epsilon_{m,k}^{(1)} = \epsilon_{m,k}^{(0)}. \quad \text{(G2)}$$

If we instead project $\langle m, k'^{(0)} | \sigma^z$ with $k \neq k'$ onto Eq. (G1), we obtain the $\mathcal{O}(\alpha)$ corrections to the eigenvectors,

$$
\left| 0, \mathrm{br.}^{(1)} \right\rangle = \frac{1}{2} \left\langle 0, \mathrm{br.}^{(0)} \middle| \sigma^x \middle| 0, \mathrm{br.}^{(0)} \right\rangle \sigma^x \left| 0, \mathrm{br.}^{(0)} \right\rangle + \int_0^\infty k\,dk\, \frac{\epsilon_{\mathrm{br.}}^{(0)}}{\epsilon_k^{(0)} - \epsilon_{\mathrm{br.}}^{(0)}} \left\langle 0, k^{(0)} \middle| 0, \mathrm{br.}^{(0)} \right\rangle \left| 0, k^{(0)} \right\rangle
$$

$$
+ \int_0^\infty k\,dk\, \frac{\epsilon_{\mathrm{br.}}^{(0)}}{\epsilon_k^{(0)} + \epsilon_{\mathrm{br.}}^{(0)}} \left\langle 0, k^{(0)} \middle| \sigma^x \middle| 0, \mathrm{br.}^{(0)} \right\rangle \sigma^x \left| 0, k^{(0)} \right\rangle,
$$

$$
\left| \pm 1, \mathrm{trans.}^{(1)} \right\rangle = 0,
$$

$$
\left| 0, k^{(1)} \right\rangle = \int_0^\infty k'\,dk'\, \frac{\epsilon_k^{(0)}}{\epsilon_k^{(0)} + \epsilon_{k'}^{(0)}} \left\langle 0, k' \middle| \sigma^x \middle| 0, k \right\rangle \sigma^x \left| 0, k' \right\rangle
$$

$$
+ \frac{\epsilon_k^{(0)}}{\epsilon_{\mathrm{br.}}^{(0)} - \epsilon_k^{(0)}} \left\langle 0, \mathrm{br.}^{(0)} \middle| 0, k^{(0)} \right\rangle \left| 0, \mathrm{br.}^{(0)} \right\rangle + \frac{\epsilon_k^{(0)}}{\epsilon_{\mathrm{br.}}^{(0)} + \epsilon_k^{(0)}} \left\langle 0, \mathrm{br.}^{(0)} \middle| \sigma^x \middle| 0, k^{(0)} \right\rangle \sigma^x \left| 0, \mathrm{br.}^{(0)} \right\rangle, \tag{G3}
$$

$$
\left| \pm 1, k^{(1)} \right\rangle = \int_0^\infty k'\,dk'\, \frac{\epsilon_k^{(0)}}{\epsilon_k^{(0)} + \epsilon_{k'}^{(0)}} \left\langle \mp 1, k'^{(0)} \middle| \sigma^x \middle| \pm 1, k^{(0)} \right\rangle \sigma^x \left| \mp 1, k'^{(0)} \right\rangle
$$

$$
\mp \left\langle \pm 1, \mathrm{trans.}^{(0)} \middle| \pm 1, k^{(0)} \right\rangle \left| \pm 1, \mathrm{trans.}^{(0)} \right\rangle.
$$

The first order corrections to the steady state coefficients in Eq. (34) are given by

$$
c_{m,\mathrm{bd.},\pm\omega}^{(1)} = c_{m,\mathrm{bd.},\pm\omega}^{(0)} \left\langle m, \mathrm{bd.}^{(0)} \middle| -m, \mathrm{bd.}^{(0)} \right\rangle
$$

$$
- c_{-m,\mathrm{bd.},\mp\omega}^{(0)} \left( \left\langle m, \mathrm{bd.}^{(0)} \middle| \sigma^x \middle| -m, \mathrm{bd.}^{(0)} \right\rangle - \left\langle m, \mathrm{bd.}^{(0)} \middle| \sigma^z \sigma^x \middle| -m, \mathrm{bd.}^{(1)} \right\rangle \right)
$$

$$
+ \int_0^\infty k\,dk\, \left( c_{m,k,\pm\omega}^{(0)} \left( \left\langle m, \mathrm{bd.}^{(0)} \middle| m, k^{(0)} \right\rangle + \left\langle m, \mathrm{bd.}^{(0)} \middle| \sigma^z \middle| m, k^{(1)} \right\rangle \right) \right.
$$

$$
\left. - c_{-m,k,\mp\omega}^{(0)} \left( \left\langle m, \mathrm{bd.}^{(0)} \middle| \sigma^x \middle| -m, k^{(0)} \right\rangle - \left\langle m, \mathrm{bd.}^{(0)} \middle| \sigma^z \sigma^x \middle| -m, k^{(1)} \right\rangle \right) \right),
$$

$$
c_{m,k,\pm\omega}^{(1)} = c_{m,k,\pm\omega}^{(0)} \tag{G4}
$$

$$
- \int_0^\infty k'\,dk'\, c_{-m,k',\mp\omega}^{(0)} \left( \left\langle m, k^{(0)} \middle| \sigma^x \middle| -m, k'^{(0)} \right\rangle - \left\langle m, k^{(0)} \middle| \sigma^z \sigma^x \middle| -m, k'^{(1)} \right\rangle \right)
$$

$$
+ c_{m,\mathrm{br.},\pm\omega}^{(0)} \left( \left\langle m, k^{(0)} \middle| m, \mathrm{bd.}^{(0)} \right\rangle + \left\langle m, k^{(0)} \middle| \sigma^z \middle| m, \mathrm{bd.}^{(1)} \right\rangle \right)
$$

$$
- c_{-m,\mathrm{bd.},\mp\omega}^{(0)} \left( \left\langle m, k^{(0)} \middle| \sigma^x \middle| -m, \mathrm{bd.}^{(0)} \right\rangle - \left\langle m, k^{(0)} \middle| \sigma^z \sigma^x \middle| -m, \mathrm{bd.}^{(1)} \right\rangle \right).
$$

Only the $m = 0$ coefficients may be calculated using Eq. (G4) and they are given by

$$
c_{0,\mathrm{br.},\pm\omega}^{(1)} = c_{0,\mathrm{br.},\pm\omega}^{(0)} \left\langle 0, \mathrm{br.}^{(0)} \middle| 0, \mathrm{br.}^{(0)} \right\rangle - \frac{1}{2} c_{0,\mathrm{br.},\mp\omega}^{(0)} \left\langle 0, \mathrm{br.}^{(0)} \middle| \sigma^x \middle| 0, \mathrm{br.}^{(0)} \right\rangle
$$

$$
- \int_0^\infty k\,dk\, \left( \frac{\epsilon_{\mathrm{br.}}^{(0)}}{\epsilon_k^{(0)} - \epsilon_{\mathrm{br.}}^{(0)}} c_{0,k,\pm\omega}^{(0)} \left\langle 0, \mathrm{br.}^{(0)} \middle| 0, k^{(0)} \right\rangle + \frac{\epsilon_{\mathrm{br.}}^{(0)}}{\epsilon_k^{(0)} + \epsilon_{\mathrm{br.}}^{(0)}} c_{0,k,\mp\omega}^{(0)} \left\langle 0, \mathrm{br.}^{(0)} \middle| \sigma^x \middle| 0, k^{(0)} \right\rangle \right),
$$

$$
c_{0,k,\pm\omega}^{(1)} = c_{0,k,\pm\omega}^{(0)} - \int_0^\infty k'\,dk'\, \frac{\epsilon_k^{(0)}}{\epsilon_k^{(0)} + \epsilon_{k'}^{(0)}} c_{0,k',\mp\omega}^{(0)} \left\langle 0, k^{(0)} \middle| \sigma^x \middle| 0, k'^{(0)} \right\rangle \tag{G5}
$$

$$
- \frac{\epsilon_k^{(0)}}{\epsilon_{\mathrm{br.}}^{(0)} - \epsilon_k^{(0)}} c_{0,\mathrm{br.},\pm\omega}^{(0)} \left\langle 0, k^{(0)} \middle| 0, \mathrm{br.}^{(0)} \right\rangle - \frac{\epsilon_k^{(0)}}{\epsilon_{\mathrm{br.}}^{(0)} + \epsilon_k^{(0)}} c_{0,\mathrm{br.},\mp\omega}^{(0)} \left\langle 0, k^{(0)} \middle| \sigma^x \middle| 0, \mathrm{br.}^{(0)} \right\rangle.
$$

The first order corrections to the steady state coefficients in the $m = \pm 1$ sectors using the $|m = -1, k = 0\rangle$ mode method in Sec. IV.2 are given by

$$
\begin{aligned}
\tilde{c}^{*(1)}_{1,k',\mp\omega} &= \frac{1}{\epsilon_0^{(0)} + \epsilon_{k'}^{(0)}} \left( 2\epsilon_0^{(0)} \tilde{c}^{*(0)}_{1,k',\mp\omega} + \left\langle 1, k'^{(0)} \middle| \sigma^x (\epsilon_0^{(0)} \sigma^z - H_{-1}\sigma^z) \middle| \begin{pmatrix} f_{-1,\pm\omega} \\ -f^*_{1,\mp\omega} \end{pmatrix} \right\rangle \right. \\
&\quad \left. - \epsilon_{k'}^{(0)} \int_{k>0}^{\infty} k\,dk\, \frac{\epsilon_0^{(0)} - \epsilon_k^{(0)}}{\epsilon_k^{(0)} + \epsilon_{k'}^{(0)}} \tilde{c}^{(0)}_{-1,k,\pm\omega} \left\langle 1, k'^{(0)} \middle| \sigma^x \middle| -1, k^{(0)} \right\rangle \right), \\
\tilde{c}^{(1)}_{-1,k',\pm\omega} &= \frac{1}{\epsilon_{k'}^{(0)} - \epsilon_0^{(0)}} \left( \left\langle -1, k'^{(0)} \middle| (\epsilon_0^{(0)} \sigma^z - H_{-1}\sigma^z) \middle| \begin{pmatrix} f_{-1,\pm\omega} \\ -f^*_{1,\mp\omega} \end{pmatrix} \right\rangle \right. \\
&\quad \left. + \epsilon_{k'}^{(0)} \int_{k>0}^{\infty} k\,dk\, \frac{\epsilon_k^{(0)} + \epsilon_0^{(0)}}{\epsilon_k^{(0)} + \epsilon_{k'}^{(0)}} \tilde{c}^{*(0)}_{1,k,\mp\omega} \left\langle -1, k'^{(0)} \middle| \sigma^x \middle| 1, k^{(0)} \right\rangle \right), \\
\tilde{c}^{(1)}_{1,\text{trans.},\mp\omega} &= 0.
\end{aligned}
\tag{G6}
$$

## Appendix H: Calculation of $\tilde{F}_i$ for the skyrmion

The neatest thing to do is to calculate $\tilde{F}_x + i\tilde{F}_y$, which is

$$
\begin{aligned}
\tilde{F}^x_{\text{trans}} + i\tilde{F}^y_{\text{trans}} = -\Bigg\langle \int & r\,dr\,d\chi\, e^{i\chi} \Bigg( \\
& \frac{1}{\sqrt{2}} \left( \theta'_0(a^{(1)} + a^{*(1)}) + \frac{\sin(\theta_0)}{r}(a^{(1)} - a^{*(1)}) \right) \Bigg( \\
& - \frac{1}{\sqrt{2}r^2} \Big( 2i(r\cos(\theta_0) + \sin(\theta_0))(\partial_\chi a^{*(1)} - \partial_\chi a^{(1)}) \\
& + 2r^2(1 + \theta'_0)(\partial_r a^{(1)} + \partial_r a^{*(1)}) \\
& + (a^{(1)} + a^{*(1)})(2r\cos(2\theta_0) + b_0 r^2 \sin(\theta_0) + \sin(2\theta_0)) \Big) \\
& + \frac{1}{2}\sin(\theta_0)\left( b_R \cos(\phi - \omega t) + b_L \cos(\phi + \omega t)\right) + b_z \cos(\theta_0)\cos(\omega t + \delta) \Bigg) \\
& + a^{*(1)}a^{(1)}\partial_r \left( 2\theta'_0 + (\theta'_0)^2 + \frac{\sin^2(\theta_0)}{r^2} + \frac{\sin(2\theta_0)}{r} - b_0\cos(\theta_0) \right) \Bigg) \Bigg\rangle_t
\end{aligned}
\tag{H1}
$$

in terms of the first order $a^{(1)}, a^{*(1)}$ fields for the Bloch/Néel skyrmion. Then $\tilde{F}^x_{\text{trans}}, \tilde{F}^y_{\text{trans}}$ are the real and imaginary parts of Eq. (H1), respectively. Notice how much more algebraically complex the $\tilde{F}^{x,y}_{\text{trans}}$ components are, compared to $F^{x,y}_{\text{trans}}$ given in Eq. (K2)!

## Appendix I: Contour integral for far-field linear response

Here, we derive $a^{(1),\text{scatt}}_{0,\pm\omega}$, with the understanding that $a^{(1),\text{scatt}}_{\pm 1,\pm\omega}$ can be obtained straightforwardly using the same technique. For $\gamma < 0$, the denominator of the $|m, k^{(0)}\rangle_u$ term in Eq. (33) reads

$$
\mp\omega - E_k = \mp\omega + (b_0 + k^2)(1 + i\alpha).
$$

The case $+\omega$ produces very strongly damped and thus physically irrelevant roots, so we can immediately set $a^{(1),\text{scatt}}_{0,-\omega}$ to zero. For the case $-\omega$, we have two complex roots $\pm k^*$, with

$$
k_\omega = k_0 - i\alpha\omega/v_g + \mathcal{O}(\alpha^2),
\tag{I1}
$$

where $k_0 = \sqrt{\omega - b_0}$, $v_g = 2k_0$. Using the far-field limits

$$
J_m(kr) = \sqrt{\frac{2}{\pi kr}} \cos\left( kr - \frac{m\pi}{2} - \frac{\pi}{4} \right),
$$

$$
Y_m(kr) = \sqrt{\frac{2}{\pi kr}} \sin\left( kr - \frac{m\pi}{2} - \frac{\pi}{4} \right),
$$

valid for $r \gg 2\pi/k$, we can write $a_{0,\omega}^{(1),\text{scatt}}$ to leading order in $\alpha$ as

$$a_{0,\omega}^{(1),\text{scatt}} = \int_{-\infty}^{\infty} \frac{-ik\,dk\,c_{k,\omega}^{(0)}}{(k-k_\omega)(k+k_\omega)} \sqrt{\frac{1}{2\pi|k|r}} \cdot \left( e^{i\left(\frac{\pi}{4}-kr-\delta_{0,k}\right)} - e^{-i\left(\frac{\pi}{4}-kr-\delta_{0,k}\right)} \right),$$

(I2)

where we define $c_{k,\omega}^{(0)} = 0$ for $k < 0$. We can convert this to an integral in the complex $k$-plane, where the integration contour is a semi-circle in the lower/upper half-plane, depending on the sign in $e^{\mp ikr}$. Applying the residue theorem, we obtain Eq. (41).

**Appendix J: Vanishing $\int \mathrm{d}^2r\,\hat{\mathbf{M}}^{(0)} \cdot \langle \dot{\mathbf{M}}^{(1)} \times \nabla_i \mathbf{M}^{(1)} \rangle_t$**

We will now show why $\int \mathrm{d}^2r\,\hat{\mathbf{M}}^{(0)} \cdot \langle \dot{\mathbf{M}}^{(1)} \times \nabla_i \mathbf{M}^{(1)} \rangle_t$ vanishes in the case of a driven single skyrmion. In polar coordinates, and taking the linear combination $\dot{P}_\perp^{\mathrm{m}} = \dot{P}_x^{\mathrm{m}} + i\dot{P}_y^{\mathrm{m}}$ to simplify the algebra, we have

$$\int \mathrm{d}^2r\,\langle \dot{P}_\perp^{\mathrm{m}} \rangle_t = \int \mathrm{d}^2r\,\langle \dot{P}_x^{\mathrm{m}} + i\dot{P}_y^{\mathrm{m}} \rangle_t =$$
$$\int r\,dr\,d\chi\,e^{i\chi}\left(\partial_r + \frac{i\partial_\chi}{r}\right) \sum_{\substack{m,m' \\ \omega'=\pm\omega}} \omega' a_{m,\omega'}^{(1)} a_{m',\omega'}^{*(1)} e^{i\chi(m-m')}.$$

In the above, we can use integration by parts to replace $i\partial_\chi/r \to 1/r$. Integrating the resulting expression over $\chi$, we can write the expression as

$$\int \mathrm{d}^2r\,\langle \dot{P}_\perp^{\mathrm{m}} \rangle_t = \int_0^\infty dr\,\partial_r \sum_{\substack{m,m' \\ \omega'=\pm\omega}} r\omega' a_{m,\omega'}^{(1)} a_{m',\omega'}^{*(1)} \delta_{m+1,m'}$$
$$= \left[ \sum_{\substack{m,m' \\ \omega'=\pm\omega}} r\omega' a_{m,\omega'}^{(1)} a_{m',\omega'}^{*(1)} \delta_{m+1,m'} \right]_0^\infty.$$

The only contribution in the $a^{(1)}$ fields which survives as $r \to \infty$ is $a_{-1,\pm\omega}^{(1),\text{const.}}$ — everything else decays on a length scale $l \sim 1/\alpha$, and therefore vanishes at large enough $r$, see also Eq. (40) and (41). Due to the $\delta_{m+1,m'}$ term, $a_{-1,\pm\omega}^{(1),\text{const.}}$ will always be paired with $a_{0,\pm\omega}^{(1)}$, which vanishes for $r \to \infty$. Thus, we can conclude that $\int \mathrm{d}^2r\,\langle \dot{P}_\perp^{\mathrm{m}} \rangle_t = 0$, and consequently also $\int \mathrm{d}^2r\,\langle \dot{P}_x^{\mathrm{m}} \rangle_t = \int \mathrm{d}^2r\,\langle \dot{P}_y^{\mathrm{m}} \rangle_t = 0$, for a single skyrmion or any other texture where the topological charge is localised and embedded in a a ferromagnet.

**Appendix K: Dependence of $F_{\mathbf{trans}}$ on $\alpha$ for $\omega \leq \omega_{\mathbf{Kit.}}$**

We are interested in the behaviour of $\mathbf{F}_{\text{trans}}$ in the limit of low damping, $\alpha \to 0$, and at driving frequencies at or below the gap, $\omega \leq \omega_{\text{Kit.}}$. We have established that only the second term in Eq. (14), which is explicitly

proportional to $\alpha$, gives a non-zero contribution to $\mathbf{F}_{\text{trans}}$. Let us denote the integrand of this term as $\mathfrak{F}^{x,y}$, such that

$$\mathfrak{F}^i = \dot{\mathbf{M}}^{(1)} \cdot \nabla_i \mathbf{M}^{(1)}.$$

(K1)

In Sec. IV, we calculated $\mathbf{M}^{(1)}$ as a function of polar coordinates $r, \chi$. For this reason, it is algebraically neater to consider the linear combination $\mathfrak{F}_x + i\mathfrak{F}_y$. We will investigate what happens as a function of $\omega$ to the time- and polar angle- averaged $\mathfrak{F}_x + i\mathfrak{F}_y$, which reads

$$\langle \mathfrak{F}^x + i\mathfrak{F}^y \rangle_{\chi,t} = \left\langle e^{i\chi}\left( \dot{a}^{(1)}\partial_r a^{*(1)} + \dot{a}^{*(1)}\partial_r a^{(1)} \right.\right.$$
$$+ \frac{i}{r}(\dot{a}^{(1)}\partial_\chi a^{*(1)} + \dot{a}^{*(1)}\partial_\chi a^{(1)})$$
(K2)
$$\left.\left. + \frac{\cos(\theta_0)}{r}\left( \dot{a}^{(1)}a^{*(1)} - \dot{a}^{*(1)}a^{(1)} \right) \right) \right\rangle_{\chi,t}$$

in terms of the linear response fields $a^{(1)}$, $a^{*(1)}$. For simplicity, we also set $\delta, h = 0$, which makes all the $c_{0,k,\pm\omega}^{(0)}$, $\tilde{c}_{\pm1,k,\pm\omega}^{(0)}$ coefficients real. Alternative choices will have an incidence on the force components $F_{\text{trans}}^{x,y}$, but not on its magnitude $|\mathbf{F}_{\text{trans}}|$. If $\omega < \omega_{\text{Kit.}}$, $\omega \neq \omega_{\text{br.}}$, i.e. we are below the gap and away from the resonances, in the limit $\alpha \to 0$ all the $a_{0,\pm\omega}^{(1)}$, $a_{\pm1,\pm\omega}^{(1)}$ fields become purely real. After some algebra, we find

$$\lim_{\alpha\to 0} \langle \mathfrak{F}^x \rangle_{\chi,t}(\omega \neq \omega_{\text{br./Kit.}}) = 0$$
$$\lim_{\alpha\to 0} \langle \mathfrak{F}^y \rangle_{\chi,t}(\omega \neq \omega_{\text{br./Kit.}}) = 2\left[ a_{0,+\omega}^{(1)'}\left( a_{-1,+\omega}^{(1)} - a_{1,+\omega}^{(1)} \right) \right.$$
$$\left. - a_{0,-\omega}^{(1)'}\left( a_{-1,-\omega}^{(1)} - a_{1,-\omega}^{(1)} \right) \right]$$
$$+ \frac{2}{r}\cos(\theta_0)\left[ a_{0,+\omega}^{(1)}\left( a_{1,+\omega}^{(1)} + a_{-1,+\omega}^{(1)} \right) \right.$$
$$\left. - a_{0,-\omega}^{(1)}\left( a_{-1,-\omega}^{(1)} + a_{1,-\omega}^{(1)} \right) \right]$$

Hence, in the limit of low damping and away from the breathing and Kittel resonances, $\langle \mathfrak{F}^x \rangle_{\chi,t}$ vanishes and only the $\langle \mathfrak{F}^y \rangle_{\chi,t}$ component survives and tends to a constant finite value. Thus, we can conclude that the force acting on the skyrmion is purely in the $\mathbf{e}_y$-direction, and given by

$$\lim_{\alpha\to 0} \mathbf{F}_{\text{trans}}(\omega \neq \omega_{\text{br.}}, \omega_{\text{Kit.}}) = \alpha \mathbf{e}_y \int 2\pi r\,dr\,\langle \mathfrak{F}^y \rangle_{\chi,t}.$$

(K3)

If instead $\omega = \omega_{\text{br.}}$ exactly, then $a_{0,\pm\omega}^{(1)}$ will be purely imaginary and grow like $1/(i\alpha)$. This changes $\langle \mathfrak{F}^{x,y} \rangle_{\chi,t}$

to

$$\lim_{\alpha \to 0} \langle \mathfrak{F}^x \rangle_{\chi,t}(\omega = \omega_{\text{br.}}) = 2i\Big[ - a_{0,+\omega}^{(1)'} \big( a_{-1,+\omega}^{(1)} + a_{1,+\omega}^{(1)} \big)$$
$$+ a_{0,-\omega}^{(1)'} \big( a_{-1,-\omega}^{(1)} + a_{1,-\omega}^{(1)} \big) \Big]$$
$$+ \frac{2i}{r} \cos(\theta_0) \Big[ a_{0,+\omega}^{(1)} \big( a_{1,+\omega}^{(1)} - a_{-1,+\omega}^{(1)} \big)$$
$$+ a_{0,-\omega}^{(1)} \big( a_{-1,-\omega}^{(1)} - a_{1,-\omega}^{(1)} \big) \Big]$$
$$\lim_{\alpha \to 0} \langle \mathfrak{F}^y \rangle_{\chi,t}(\omega = \omega_{\text{br.}}) = 0.$$

This time the situation is reversed and $\langle \mathfrak{F}^y \rangle_{\chi,t}$ vanishes. The resulting force is

$$\lim_{\alpha \to 0} \mathbf{F}_{\text{trans}}(\omega = \omega_{\text{br.}}) = \alpha \mathbf{e}_x \int 2\pi r \, dr \langle \mathfrak{F}^x \rangle_{\chi,t}. \quad \text{(K4)}$$

Importantly, now we have $\langle \mathfrak{F}^x \rangle_{\chi,t} \propto 1/\alpha$ in the limit $\alpha \to 0$ because of the $a_{0,\pm\omega}^{(1)}$ components. Hence,

$$\lim_{\alpha \to 0} F_{\text{trans}}(\omega = \omega_{\text{br.}}) \sim \alpha \cdot \frac{1}{\alpha} \sim \text{const.} \quad \text{(K5)}$$

Finally, we consider the case $\omega = \omega_{\text{Kit.}} = b_0$. In this case, depending on $\text{sgn}(\gamma)$ either $a_{-1,+\omega}^{(1)}$ and $a_{1,-\omega}^{(1)}$ or $a_{1,+\omega}^{(1)}$ and $a_{-1,-\omega}^{(1)}$ will be purely imaginary and $\propto 1/(i\alpha)$, see Eq. (39). If $\gamma < 0$ and $b_R > 0$, $a_{-1,+\omega}^{(1)}$ and $a_{1,-\omega}^{(1)}$ are purely imaginary and we have

$$\lim_{\alpha \to 0} \langle \mathfrak{F}^x \rangle_{\chi,t}(\omega = \omega_{\text{Kit.}}) = 2i\Big( a_{0,+\omega}^{(1)'} a_{-1,+\omega}^{(1)} - a_{0,-\omega}^{(1)'} a_{1,-\omega}^{(1)} \Big)$$
$$+ \frac{2i}{r} \cos(\theta_0) \Big( a_{0,+\omega}^{(1)} a_{-1,+\omega}^{(1)} + a_{0,-\omega}^{(1)} a_{1,-\omega}^{(1)} \Big)$$
$$\lim_{\alpha \to 0} \langle \mathfrak{F}^y \rangle_{\chi,t}(\omega = \omega_{\text{Kit.}}) = 0. \quad \text{(K6)}$$

The forces are therefore the same as in Eq. (K4), but with $\langle \mathfrak{F}^x \rangle_{\chi,t}$ as defined in Eq. (K6). Thus at the Kittel resonance we also have

$$\lim_{\alpha \to 0} F_{\text{trans}}(\omega = \omega_{\text{Kit.}}) \sim \text{const.} \quad \text{(K7)}$$