# Peer review of "Skyrmion Jellyfish in Driven Chiral Magnets"

_SciPost Physics_

## Round 1 · Referee Report · Anonymous (Referee 1) · 2023-1-26

Report

This work is on a method for exciting a propagation mode of a skyrmion in chiral magnets. The method is based on an oscillating magnetic field that works in combination with friction or magnon emission. Eqs. (8) and (9) are the starting points for the calculations. They are assuming an expansion of the skyrmion configuration using a small parameter epsilon and we thus expect approximate results to be derived that will nevertheless demonstrate skyrmion motion.

Calculations are carried out using a Thiele-type equation and extensive results on the linear response of skyrmions, that is, on magnon modes. It is demonstrated that a skyrmion responds to the oscillating field not only by exciting oscillatory modes but also by translational motion. This is a very interesting phenomenon, explained to be related to two different mechanisms, working together with friction and magnon emission, respectively. The motion of a skyrmion is a very interesting problem both from the theoretical point of view as well as for any application that this system may have. This paper offers an extensive discussion of a non-trivial procedure that can produce skyrmion translational motion.

The mechanism is noted to resemble the method for jellyfish motion. The dynamics in a continuum system, such as a magnet, is indeed related to hydrodynamics but making the analogy precise is typically quite hard to carry out.

I have three main comments on the technical part of the paper.

1) Following Eq. (8), it is stated that the linear response is "purely oscillatory" and the next order term contains the translational motion. This is plausible, but can it be justified? Eq. (9) seems to be a guess if no further justification is given.

Also, after Eq. (9), it is stated that "the O(epsilon^2) terms grow linearly in t". This is a somewhat unclear statement because it seems to assume that Eq. (9) is more precise than Eq. (8). Could one not also argue in the opposite way, i.e., that probably Eq. (8) is valid at all times while Eq. (9) is valid only for short times?

2a) Eq. (12) is a Thiele-type equation that is used for deriving the main results. It should be mentioned that the standard Thiele equation is not valid in the initial stages of the skyrmion motion (for example, in the usual case of application of a constant field or force). This is because the derivation of the Thiele equation assumes a steady state (and this cannot be achieved instantly after the application of the force). Actually, the difference between the motion (velocity) in the initial stages and the steady state motion can be of the order of the steady state velocity. Since we have here a time-varying field, can the Thiele-type equation (12) be justified?

2b) As an additional justification of Eq. (12), it is mentioned that it can be derived using the results of Appendix C. But, in Eq. (C7), one sees that choosing bext=0 and dot-M=0 gives dot-P=0, and thus a constant momentum. Does this imply that skyrmion motion can be obtained also without the oscillating field (if one chooses constant but nonzero momentum)? Also, if damping is assumed small, the simpler condition bext=0 appears to be adequate for skyrmion motion. This would question the origin of the skyrmion motion studied in the paper.

3) Choosing to describe the motion via a Thiele-type equation, a certain position coordinate for a skyrmion is implicitly chosen (a skyrmion is an extended object and it does not have a uniquely defined position). Therefore, one could plausibly imagine that the studied oscillations may be due to not choosing the optimum coordinates. A non-optimum choice of coordinates would introduce artifacts that complicate and obscure the dynamical response. Could it be argued that the Thiele coordinates (implicitly chosen) are anything more than plausible and that they are suitable for this problem? The presentation of results does not seem to support the choice of the coordinates.

Some other comments.

1) The choice b0=1 is noted at the end of Sec. II. Would another choice for b0 not change the skyrmion size? In the literature, big skyrmions where the skyrmion domain wall can be approximated as a 1D wall, are often treated separately. Is it necessary to discuss separately cases for various values of b0?

2) b_i is defined just before Eq. (5), but only b_0 appears in it.

3) Eq. (21) is used to give Eq. (D1) and this is said to be "too long to list here". But, Eq. (D1) seems to be possible to take a relatively compact form, and thus be listed in the main text.

4) In Sec. IV.2, a technical problem is noted for the k=0 mode. It is not clear to me whether this might be an essential problem and not only a technical one. On the other hand, Eq. (39) seems to imply that this problem was indeed technical.

5) Sec. V. There is some confusion in references to Eq. (14) and Eq. (44).

6) I suggest that using sgn(gamma) complicates the expressions without much benefit. The authors could just choose a sign for gamma.

7) Given the length of the paper, it will probably be beneficial for its clarity to omit or relegate to appendix comments that are not of central importance. For example, the discussion of the helix in the paragraph containing Eq. (15) seems to be more related to another paper (Ref. [17]).

The paper gives an extensive discussion of a non-trivial procedure that can produce skyrmion translational motion. It employs a number of methods, for example, those related to magnon modes. It does have the potential to excite interest within the community. However, it is not clear whether the methods used are optimum for this problem. As a result, it is not clear whether the reported phenomenon is unambiguously demonstrated and clearly discussed. Furthermore, the application of the methods contains a number of assumptions that appear to make them specific to the current problem, and their presentation is complicated enough thus making it unlikely that they will be used in other related problems. While I think that the authors could improve the clarity of the presentation and publish their results in some form, I do not recommend the publication of this paper in SciPost as I can not see that the acceptance criteria might be satisfied.

  • validity: good
  • significance: high
  • originality: high
  • clarity: good
  • formatting: -
  • grammar: excellent

Author:  Nina del Ser  on 2023-03-26  [id 3513]

(in reply to Report 2 on 2023-01-26)
Category:
answer to question

We thank the referee for the positive report and thought-provoking comments and suggestions on how to improve the paper. Below we cite all points raised by the referee and provide a reply.

“...I have three main technical comments on the technical part of the paper”

1) “Following Eq.(8), it is stated that the linear response is “purely oscillatory” and the next order term contains the translational motion. This is plausible, but can it be justified? Eq.(9) seems to be a guess if no further justification is given.

To check why a linear in $t$ term $\it{cannot}$ appear at order $\mathcal{O}(\epsilon)$ with our choice of driving magnetic field, we can assume the contrary and see what happens when we substitute such an ansatz into the LLG. In this case, considering the $\mathcal{O}(\epsilon)$ terms of the LLG would on the one hand give us constant (in time) terms such as $\bf v_{\text{trans}}\cdot \nabla \bf M^{(0)}(\bf r)$ coming from the time derivative of $\bf M(\bf r-\epsilon \bf v_{\text{trans}} t)$, and on the other hand terms oscillating at $e^{\pm i\omega t}$ coming from the $\bf M\times\bf B_{\text{eff}}$ term. These cannot be balanced, meaning that the ansatz is wrong. If instead we assume that $\bf v_{\text{trans}}$ is to lowest order $\mathcal{O}(\epsilon^2)$, some of the $\mathcal{O}(\epsilon^2)$ terms coming from $\bf B_{\text{eff}}$ consist of products of first order terms with opposite frequencies, $e^{i\omega t} e^{-i\omega t}=1$, which are now constant in time and can balance the constant $\bf v_{\text{trans}}\cdot \nabla \bf M^{(0)}(\bf r)$ terms. We have modified the text around Eq (8) to hopefully make this more clear.

"Also, after Eq.(9), it is stated that “the $\mathcal{O}(\epsilon^2)$ terms grow linearly in t”. This is a somewhat unclear statement because it seems to assume that Eq.(9) is more precise than Eq.(8). Could one not also argue in the opposite way, i.e., that probably Eq.(8) is valid at all times while Eq.(9) is valid only for short times?”

Once the skyrmion has had enough time to “swim” to a location far enough from its starting position, the spin configuration cannot be accurately described by the naïve perturbative expansion given in Eq.(8). The correct expansion which solves this problem and is valid at all times, up to and including $\mathcal{O}(\epsilon^2)$, is Eq.(9).

2)a) “Eq. (12) is a Thiele-type equation that is used for deriving the main results. It should be mentioned that the standard Thiele equation is not valid in the initial stages of the skyrmion motion (for example, in the usual case of application of a constant field or force). This is because the derivation of the Thiele equation assumes a steady state (and this cannot be achieved instantly after the application of the force). Actually, the difference between the motion (velocity) in the initial stages and the steady state motion can be of the order of the steady state velocity. Since we have here a time-varying field, can the Thiele-type equation (12) be justified?"

Indeed, the Thiele equation is only valid after the initial transient responses have decayed and we have entered the steady state. Formally, the Thiele equation (or, more precisely a variant which also includes noise terms, see Ref. [25] in the new version of the paper) becomes exact on time scales which are much larger than all other characteristic frequencies of the system, including the frequency of the oscillating system. Using the slang of “effective field theories”, one could say that we have “integrated out” all the rapidly fluctuating components and derived the equation of motion for the only remaining degree of freedom, which is the skyrmion coordinate. Thus, we claim that our results are exact in the low-frequency limit up to second order in the oscillating field, as corroborated by our comparison to numerics. We have added remarks on the validity of our effective Thiele equation at the end of Sec.III.

2)b) “As an additional justification of Eq.(12), it is mentioned that it can be derived using the results of Appendix C. But, in Eq.(C7), one sees that choosing $\bf b_{\text{ext}}=0$ and $\dot{\bf M}=0$ gives $\dot{\bf P}=0$, and thus a constant momentum. Does this imply that skyrmion motion can be obtained also without the oscillating field (if one chooses constant but nonzero momentum)? Also, if damping is assumed small, the simpler condition $\bf b_{\text{ext}}=0$ appears to be adequate for skyrmion motion. This would question the origin of the skyrmion motion studied in the paper.”

$\bf P$ in Eq. C7 is the momentum density of the full system (including not just the skyrmion, but also the magnons). When integrated over space one obtains the total momentum, which stays constant over time if the magnetization is time-independent. For a moving skyrmion, however, the magnetisation is not time-independent and $\bf P$ is not conserved due to the damping term.

The referee also asks what happens when the damping can be completely neglected, $\alpha=0$. In this case the total momentum is conserved within the continuum model we used. But even in this case, there can be momentum transfer from the skyrmion, e.g. to thermally excited magnons.

Finally, one can consider the case where there are no thermal magnons and there is no damping. In this case the skyrmion momentum is conserved. However, this does not imply that the skyrmion has a finite velocity. The skyrmion behaves like a particle in a magnetic field where $\bf P$ is identified with the canonical momentum. Thus the $x$-component of the canonical skyrmion momentum $P_x$ is actually proportional to the $y$-component of the skyrmion position $R_y$, see Ref. [17]. Momentum conservation therefore implies that even in this case, the skyrmion is not moving.

3) “Choosing to describe the motion via a Thiele-type equation, a certain position coordinate for a skyrmion is implicitly chosen (a skyrmion is an extended object and it does not have a uniquely defined position). Therefore, one could plausibly imagine that the studied oscillations may be due to not choosing the optimum coordinates. A non-optimum choice of coordinates would introduce artifacts that complicate and obscure the dynamical response. Could it be argued that the Thiele coordinates (implicitly chosen) are anything more than plausible and that they are suitable for this problem? The presentation of results does not seem to support the choice of the coordinates.”

The precise choice of the skyrmion coordinate is indeed very important when one discusses its short-time dynamics, see point 2a above. However, it does not matter for the long-time dynamics. The finite velocity of the skyrmion is independent of any reasonable choice of the skyrmion coordinate. The revised version includes a description of how we choose the skyrmion coordinate in our numerics (see last paragraph of Sec. V).

"Some other comments"

1) “The choice $b_0=1$ is noted at the end of Sec. II. Would another choice for $b_0$ not change the skyrmion size? In the literature, big skyrmions where the skyrmion domain wall can be approximated as a 1D wall, are often treated separately. Is it necessary to discuss separately cases for various values of $b_0$?”

Indeed, another choice of $b_0 $would change the skyrmion size and consequently shift the breathing eigenfrequency, as well as the gap in the magnon spectrum. But, if the skyrmion remains a skyrmion (i.e. $b_0$ is large enough so that it does not undergo a phase transition to some other lower energy magnetic texture such as a meron or skyrmion lattice), we would not expect the qualitative results to change in any way. There is a limit to this claim if we concentrate purely on numerical simulations – namely, if the size of the skyrmion becomes comparable to the system size. In this case, edge effects become important and will affect the dynamics.

One physical feature which we have not considered but which is expected to become more important in the limit of large skyrmion radii is magnon-magnon scattering. If the skyrmion radius becomes so large that the emitted magnons scatter before they make it out of the skyrmion, this will hamper the linear motion and reduce the velocity $v_{\text{trans}}$. We could include this effect via a hydrodynamic equation of motion for the magnons, but this is beyond the scope of the current paper.

2) “$b_i$ is defined just before Eq.(5), but only $b_0$ appears in it.”

We were trying to say that $b_i$ denotes the rescaled physical $B_i $ (constant and oscillating) field components, but we agree that the notation was confusing. For this reason we changed the text above Eq.(5) to just refer to the static component $B_0 $of the external magnetic field. Then we introduced the rescaled oscillating field components $b_i=(M_0J/D^2)B_1^i$, $i=x,y,z$ which appear in Sec. IV in App. E.

3) “Eq.(21) is used to give Eq.(D1) and this is said to be “too long to list here”. But, Eq.(D1) seems to be possible to take a relatively compact form, and thus be listed in the main text.“

We relegated Eq.(D1), which is the full non-linearised equation of motion for the bosonic $a$ fields, to an appendix as this is not the equation we actually solved in any of our calculations. Instead we use the linearised in $a$ version of Eq.(D1), given in Eq.(22) in the main text. This approximation works very well in the limit of weak oscillating fields in which we are working. We have modified the text in the paper to explain this reasoning.

4) “In Sec. IV.2, a technical problem is noted for the $k=0$ mode. It is not clear to me whether this might be an essential problem and not only a technical one. On the other hand, Eq.(39) seems to imply that this problem was indeed technical.”

The referee is right: this is a technical problem closely related to a physical feature. We drive the system with a homogenous oscillating field, thereby exciting the $k=0$ magnon in the ferromagnetic bulk. This effect must be included in our calculation. As this is not so easy within the scattering formalism (which assumes a static state at infinity), we have developed an alternative approach. The revised version makes this point clearer (see first two paragraphs of Sec. IV.2).

5) “Sec. V. There is some confusion in references to Eq.(14) and Eq.(44).”

Thanks for pointing out these typos, which have now been corrected.

6) “I suggest that using $\text{sgn}(\gamma)$ complicates the expressions without much benefit. The authors could just choose a sign for $\gamma$.”

In the past we have had much confusion with different definitions of $\gamma$. In particular, many introductory texts (e.g., see the Wikipedia article on Larmor precession) define $\gamma$ with the opposite convention to the one used in much of the skyrmion literature. As we wanted to make the article accessible to everyone, we took the trouble to use a convention where the sign of $\gamma$ could be toggled between positive and negative.

7) “Given the length of the paper, it will probably be beneficial for its clarity to omit or relegate to appendix comments that are not of central importance. For example, the discussion of the helix in the paragraph containing Eq.(15) seems to be more related to another paper (Ref.[17]).”

We think the discussion of the helix is useful as it demonstrates the application of the formalism we developed (valid for $\textit{any}$ magnetic texture in 1,2 or 3D) to the simplest possible one-dimensional case of a helical/conical texture. In Ref.[20] (Ref.[17] in the old version of the paper) the calculation is done in a different, more roundabout way (albeit with the exact same result), thus we thought a short discussion would be useful here.

---

## Round 1 · Referee Report · Anonymous (Referee 2) · 2023-3-3

Report

My review for the manuscript "Skyrmion Jellyfish in Driven Chiral Magnets" follows:

  • The work is extensive and thorough using a rich approach and it should be of high interest in the very active field of magnetic skyrmionic textures. It should be of relevance both to experimentalists and theorists. The motion of magnetic skyrmions is also very interesting for novel technological applications that aim to use skyrmionic textures as information carriers, e.g. in applications of nanocomputing/neuromorphic computing.

  • The authors describe the motion response of a single magnetic skyrmion to a (weak) oscillating magnetic field and how the mechanism underlying this motion is similar to a jellyfish swimming through water.

  • The authors show that the skyrmion moves as a result of periodic and asymmetric deformations in its shape.

  • The analytical theory is inspired by the Thiele equation and can be used to find the translational velocity for a magnetic texture.

  • The authors realise what they call a skyrmion "jellyfish" by driving a chiral magnet with oscillating magnetic fields.

  • The authors discuss the case where they drive a chiral magnet with weak oscillating magnetic fields to active the translational modes but also discuss what happens if they increase the driving field B1(t) strength.

  • The authors use their analytical approach to make experimental predictions about the translational speed Vtrans of a driven skyrmion for MnSi. Since the derived velocity is larger than experimentally reported depinning velocities. I find this very interesting and important.

  • Their analytical predictions match Micromagnetics simulations (this is also important as this is the standard in the field).

  • With regards to the journal's general acceptance criteria, I think it satisfies all of them as i) it is clearly explained (although it is a challenging topic/methodology), ii) the abstract and the introduction provide clear context and summary of achievements, iii) it is sufficiently detailed, iv) the citations are mostly representative and complete, iv) it contains a clear summary.

  • According to the stated SciPost's "Expectations" criteria, it is not clear to me that this work directly fits fully under one of the categories. That said, I think it is a fresh approach and significant work. I would still recommend it (in the light of my previous comment about the expectations categories) for publication with minor corrections.

Requested changes

  • Whilst I think that the citations are thorough and complete, the authors may want to consider the possible relevance of i) I Makhfudz, B Krüger, O Tchernyshyov, Physical review letters 109 (21), 217201 (theory) and ii) S. L. Zhang, W. W. Wang, D. M. Burn, H. Peng, H. Berger, A. Bauer, C. Pfleiderer, G. van der Laan & T. Hesjedal Nature Communications volume 9, Article number: 2115 (2018) (experimental).

  • With regards to the discussion that the skyrmion moves as a result of periodic and asymmetric deformations in its shape, do the authors think that it might be useful to make a link with "inertia" (as in their Ref. [7])?

  • Perhaps the authors would want to comment more about how their approach would also be relevant for other skyrmionic textures of different spin complexity like i) the topologically trivial skyrmionium or ii) skyrmion bubbles in systems with zero DMi (since they cite such cases. Especially since they explicitly mention in the manuscript that their analytical approach is able to predict the resulting translational velocity for any magnetic texture.

  • validity: high
  • significance: high
  • originality: high
  • clarity: high
  • formatting: excellent
  • grammar: excellent

Author:  Nina del Ser  on 2023-03-26  [id 3514]

(in reply to Report 1 on 2023-03-03)

We thank the referee for the positive report, as well as for the helpful comments and suggestions given to improve the paper. Below we list these points and provide our replies and the modifications carried out to address these issues.

1) “Whilst I think that the citations are thorough and complete, the authors may want to consider the possible relevance of i) Makhfuds, B Krüger, O Tchernyshyov, Physical review letters 109 (21), 127201 (theory) and ii)S.L. Zhang, W.W.Wang, D.M. Burn, H. Peng, H. Berger, A. Bauer, C. Pfleierer, G. van der Laan & T. Hesjedal Nature Communications volume 9, Article number:2115 (2018)(experimental)”

Thanks for these references, which have been added to the text. We also added “Schütte, Iwasaki, Rosch and Nagaosa, Physical Review B 90, 174434 (2014)” for the discussion of inertia as well as “Zhang, Shilei and Kronast, Florian and van der Laan, Gerrit and Hesjedal, Thorsten Nano letters Vol 18 (2018)” to go with the additional discussion of skyrmionium (see point 3 below).

2) “With regards to the discussion that the skyrmion moves as a result of periodic and asymmetric deformations in its shape, do the authors think that it might be useful to make a link with “inertia” (as in their Ref.[7])?

As pointed out by the referee, the mass of the skyrmion arises from deformation of the skyrmion when it is moving. Nevertheless, we think that very little relation between mass and the skyrmion velocity $\bf v_{\text{trans}}$ calculated in the paper. First, it is important to point that the deformation of the skyrmion in our setting occurs on frequency scales where magnons exist. At this frequency scale, the Thiele equation is not valid and one cannot even define a skyrmion mass. The Thiele equation is, however, valid at very long times, where one can also define mass terms. But in this regime, our paper only focusses on the steady-state motion with constant velocity, where the mass term does not contribute, as $\ddot{{\bf R}}=\dot{{\bf v}}_{\text{trans}}=0$. This discussion has been added to the paper in the last paragraph of Sec III.

3) “Perhaps the authors would want to comment more about how their approach would also be relevant for other skyrmionics textures of different spin complexity like i) the topologically trivial skyrmionium or ii) skyrmion bubbles in systems with zero DMi (since they cite such cases. Especially since they explicitly mention in the manuscript that their analytical approach is able to predict the resulting translational velocity for any magnetic texture.)

This is a good point and we have revised the text in the second paragraph of the conclusion to also include a discussion of skyrmionium, skyrmion bubbles and the speed enhancement that could be gained by driving the skyrmion near a wall. While we expect vtrans to be of a similar order for skyrmion bubbles, for the topologically trivial skyrmionium we would expect a $1/\alpha$ enhancement to $v_{\text{trans}}$ coming from the dissipation term. This can also be achieved by driving a skyrmion near a wall (as the wall compensates the gyrocoupling force ${\bf G}\times {\bf v}_{\text{trans}}$).

---

## Editorial Decision

resubmitted